# VisMin: Visual Minimal-Change Understanding

**Rabiul Awal**[*]   **Saba Ahmadi**[*]   **Le Zhang**[*]   **Aishwarya Agrawal**
Mila - Quebec AI Institute
Université de Montréal
{rabiul.awal,le.zhang,aishwarya.agrawal}@mila.quebec

## Abstract

Fine-grained understanding of objects, attributes, and relationships between objects is crucial for visual-language models (VLMs). To evaluate VLMs' fine-grained understanding, existing benchmarks primarily focus on evaluating VLMs' capability to distinguish between two very similar *captions* given an image. In this paper, our focus is on evaluating VLMs' capability to distinguish between two very similar *images* give a caption. To this end, we introduce a new, challenging benchmark termed **Vis**ual **Min**imal-Change Understanding (VisMin), which requires models to predict the correct image-caption match given two images and two captions. Importantly, the image pair (as well as the caption pair) contains minimal-changes, i.e., between the two images (as well as between the two captions), only one aspect changes at a time from among the following possible types of changes: *object*, *attribute*, *count*, and *spatial relation*. These four types of minimal-changes are specifically designed to test the models' understanding of objects, attributes of objects (such as color, material, shape), counts of objects and spatial relationship between objects. To curate our benchmark, we built an automatic framework using large language models and diffusion models, followed by a rigorous 4-step verification process by human annotators. Empirical experiments reveal that current VLMs exhibit notable deficiencies in understanding spatial relationships and counting abilities. Furthermore, leveraging the automated nature of our data creation process, we generate a large-scale training dataset, which we use to finetune CLIP (a foundational VLM) and Idefics2 (a multimodal large language model). Our findings show that both these models benefit significantly from fine-tuning on this data, as evident by marked improvements in fine-grained understanding across a wide range of benchmarks. Additionally, such fine-tuning improves CLIP's general image-text alignment capabilities too. We release all resources including the benchmark, the training data and the finetuned model checkpoints at `https://vismin.net/`.

## 1   Introduction

Fine-grained understanding of objects, attributes, and their relationships is critical for Visual-Language Models (VLMs) to generalize effectively to new, unseen scenes and compositions. Previous studies such as ARO [43] and Sugarcrepe [8], highlighting the deficiencies of VLMs in this domain predominantly focus on understanding fine-grained differences between two very similar *captions* – a human-written caption and an automatically generated hard-negative[2] caption, where the hard-negative caption differs from the original caption only with respect to an *object*, or an *attribute* or a *relationship* between two objects. While such hard-negative examples for *captions* can be synthesized using rule-based approaches, synthesizing such hard-negative examples for images is very

---

[*]denotes equal contribution

[2]In the context of contrastive learning, a hard-negative is a specific type of negative example that is particularly challenging to distinguish from the positive example.

38th Conference on Neural Information Processing Systems (NeurIPS 2024).

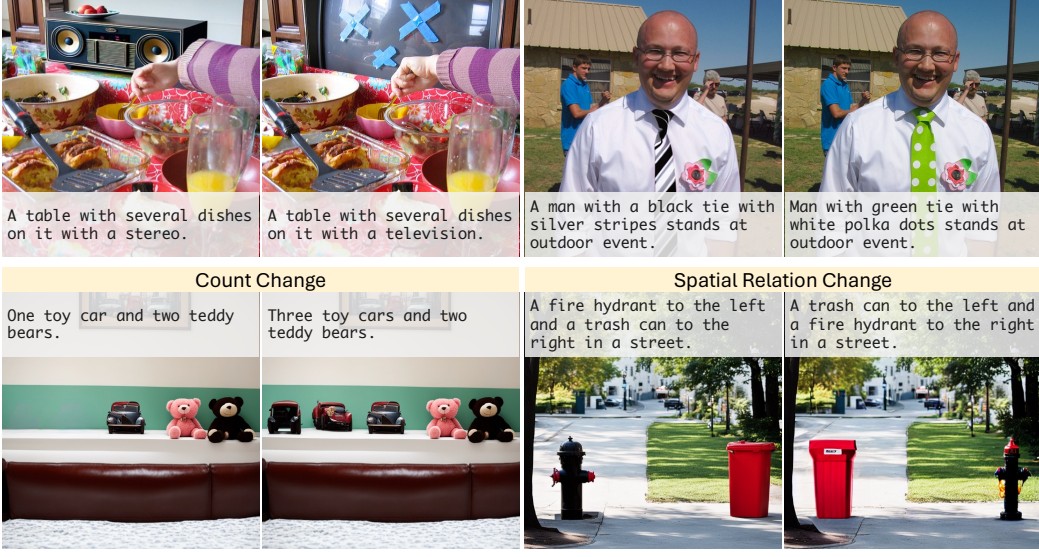

Figure 1: Overview of our VisMin benchmark. VisMin consists of four types of minimal-changes – object, attribute, count and spatial relation – between two image-captions pairs. The evaluation task requires a model to predict the correct image-caption match given: 1) two images and one caption, 2) two captions and one image.

challenging. Existing benchmarks presenting *visual* hard-negatives suffer from two main limitations: 1) **Limited Difficulty:** In benchmarks such as Winoground [36], MMVP[37], the original images and their hard-negative counterparts differ in multiple aspects (objects, attributes of objects, image background, etc.). This multiplicity limits the difficulty of the benchmark and makes it challenging to precisely evaluate the models' fine-grained understanding of specific aspects. 2) **Limited Complexity:** Although benchmarks such as EQBEN [38], SPEC [28] have controlled hard-negatives, the visual domain is limited to graphic engines, a few video domains or reliance on purely synthetic images depicting simplistic scenes.

Motivated by these observations, we propose a new benchmark, **Vis**ual **Min**imal-Change Understanding (VisMin ), built on top of the images from the COCO [22] dataset that consists of complex everyday scene images. VisMin is designed to measure VLMs' ability to comprehend minimal changes, i.e., changes with respect to only one aspect (see Fig. 1), from among the following aspects: *object*, *attribute*, *count*, and *spatial relation*, while keeping other aspects unchanged as much as possible.

The evaluation task for a model is to predict the correct image-caption match given: 1) two images and one caption, 2) two captions and one image. To curate VisMin , we built an automated pipeline using large language models and diffusion models. To ensure the quality of our benchmark, the synthetic data generated using the automated pipeline undergoes a rigorous 4-step verification process by human annotators, with data retained in the benchmark only if it passes all four steps. We meticulously designed the benchmark ensuring uniformity across various categories to the extent possible. We conduct a detailed analysis of our benchmark, which enables a more transparent assessment of the various strengths and weaknesses of the models.

We conducted empirical tests on eight open-source VLMs, including foundational models like CLIP [31] and Multimodal Large Language Models (MLLMs) such as Llava[24] and Idefics2[15]. We also evaluated two closed-source APIs, GPT-4 and Gemini. Our findings suggest that both foundational models and MLLMs perform relatively well in understanding minimal changes in objects and attributes. Surprisingly, MLLMs underperform foundational VLMs in object and attribute understanding! For spatial relation understanding, although MLLMs perform better than VLMs, both families of models perform below random chance! Similarly, both families of models show considerable room for improvement in counting capabilities. Our results underscore the need for a emphasis on spatial reasoning and counting understanding over attribute/object recognition in VLM evaluations. We anticipate that our benchmark will catalyze advancements in these critical areas within the community.

Lastly, owing to the automated nature of our synthetic data creation process, we generated a large-scale (64,392 samples) minimal-change image-text data for fine-tuning the VLMs to enhance their fine-grained understanding. Fine-tuning CLIP (a foundational VLM) and Idefics2 (a MLLM) on our minimal-change data, without any additional modifications to the model architecture or loss functions, results in significant improvements in fine-grained understanding across various benchmarks. Notably, such fine-tuning also enhances foundational VLMs' general image-text alignment capabilities as evident by marked improvements in CLIP's image-text retrieval performance on COCO. These observations suggest that our minimal-change dataset can serve as model-agnostic, general-purpose resource to enhance the capabilities of VLMs.

To summarize, our contributions are threefold: 1) **A controlled and challenging benchmark**. We introduce the VisMin benchmark, which challenges models to detect semantic differences between visually similar but semantically different images. Extensive testing on foundational VLMs and MLLMs reveals their difficulties with this task, highlighting areas for improvement. 2) **A pipeline for automated data creation and benchmark development**. We create an automated pipeline to generate visual minimal-change data at scale using large language models and diffusion models, with a rigorous four-step human verification system to ensure high data quality. 3) **Enhancement of VLMs' fine-grained understanding with fine-tuning on minimal-change data**. We improve the fine-grained understanding of CLIP and Idefics2 by fine-tuning them on our large-scale minimal-change image-text data, demonstrating improved image-text alignment and overall performance.

## 2   Related work

**Fine-grained understanding benchmarks:** Most existing benchmarks focus on understanding fine-grained textual differences, such as VL-checklist [47], ARO [43], and Sugarcrepe [8]. Benchmarks presenting visual hard-negatives, such as EQBEN [38], Winoground [36], ImageCode [14], SPEC [28], either lack minimal changes or have limited visual complexity – graphic engines, a few video domains or purely synthetic images depicting simplistic scenes. Our benchmark addresses these gaps by utilizing the advances in LLMs [10] and diffusion models [30, 20, 21] to achieve minimal changes in complex COCO-like scenes without compromising the naturalness of the images, thus providing a more robust evaluation of fine-grained visual understanding in VLMs. Detailed comparisons of benchmarks are provided in section 4.

**Automatic approach to generate visual hard negatives:** Existing approaches to automatically generate visual hard negatives fall into three broad categories: (i) using nearby video frames with semantic changes [14, 38], (ii) using graphic engines [38], (iii) using diffusion models [28, 38, 17]. Our proposed framework falls in the third category. DEMON[17] is the closest to our work, creating training data using diffusion models to improve the learning of a given vision-language model. They use diffusion models to perform local editing on the images given the target object mask. However, this approach requires attention masks from the vision-language model being studied. SPEC [28] proposes a diffusion-based canvas-filling method for generating minimally-different image pairs limited to four types of minimal changes: size, position, count and existence. Compared to these existing methods, our automated pipeline to generate minimal-change data is more involved in order to achieve minimal-changes in complex scenes while maintaining the photo-realism of the scene and controlling changes across diverse categories. Our pipeline also has more a comprehensive automated filtering mechanism compared to previous pipelines that mainly rely on CLIP-based filtering.

**Enhancing fine-grained understanding in VLMs with hard negatives:** Most methods to enhance fine-grained understanding in VLMs like CLIP focus on fine-tuning with caption-based hard negatives and optimizing loss functions to better use these signals [42, 46, 33]. Common strategies for generating textual hard negatives include heuristic rules [43], language models [46, 5], scene-graph information [7, 33], and LLMs integrated with semantic segmentation [6]. In contrast, fewer works explore visual hard negatives; methods like NegCLIP [42] and General Scene Difference [16] rely on nearest-neighbor images, which often differ too much or too little in context, limiting fine-grained learning. Our approach is closest to SPEC [28] and CounterCurate [45], which also fine-tune VLMs using minimal-change visual hard negatives. Unlike SPEC, we and CounterCurate extend this to multimodal large language models, but our work evaluates performance on 10 out-of-distribution benchmarks (compared to 1 or 2 in SPEC and CounterCurate) and outperforms baseline models in most cases, demonstrating the strength of our approach (see Tables 3 and 4).

# 3 Minimal-Change Image-Text Dataset Creation

We devised a framework to synthesize large-scale minimal-change data and introduce the VisMin benchmark (see overview fig. 2). The pipeline includes three stages: **Minimal-Change Pairs Synthesis**, where we minimally edit image and text pairs; **Automatic Filtering**, which verifies the faithfulness of the texts and synthesized images; and **Human Verification**, a four-step process to ensure that only data meeting all quality criteria is included. We will discuss each stage in detail.

Figure 2: Our dataset creation pipeline includes three stages: (i) **Minimal-Change Pairs Synthesis**: We develop methods for synthesizing minimal-change image-caption pairs involving Objects & Attributes and Counting & Spatial Relations. (ii) **Automatic Filtering**: An LLM generates questions and answers based on captions, and a VQA model predicts answers from images. Synthetically generated minimal-change data are excluded if answers don't match. (iii) **Human Verification**: Synthetically generated minimal-change data undergoes a rigorous 4-steps human verification, and only examples passing all stages are included in the benchmark.

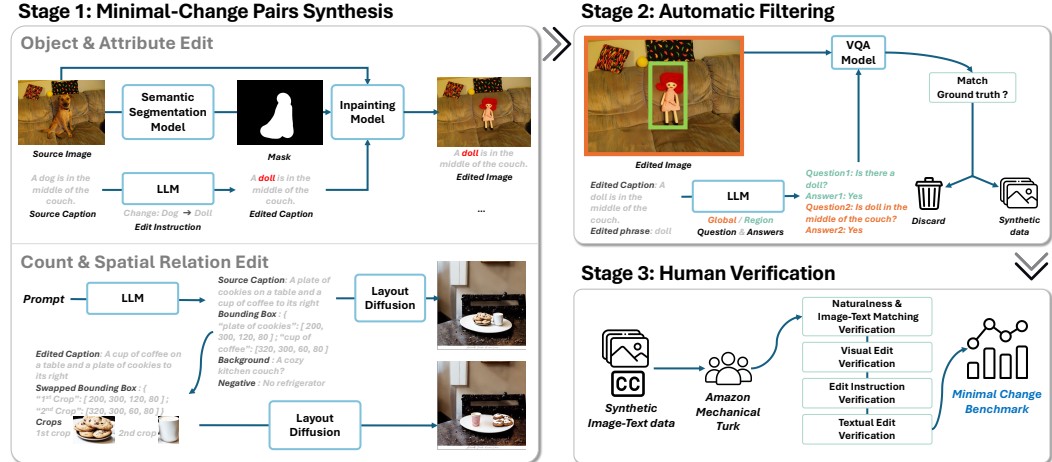

**Stage 1: Minimal-Change Pairs Synthesis** In the first stage of our pipeline, we focus on synthesizing minimal-change image-text pairs across four strategic categories: objects, attributes, counting, and spatial relations. These categories are specifically chosen to test various levels of visual-linguistic comprehension. We generate minimal-change text pairs using LLM and then generate minimal-change image pairs using diffusion models. Our synthesis process distinctly tailors the creation of image-text pairs to the specific needs of each category, depicted in Stage 1 in Figure 2 (Object & Attribute Edit and Count & Spatial Relation Edit blocks).

**LLM-guided Edit Instructions Generation** To generate minimal-change text pairs, we start with source captions and then prompt an LLM (Mistral 47B [10]) to generate both the edit instructions specific to each edit category and the corresponding edited caption (see Appendix A.2.1 for the prompt used). For **Object and Attribute** edits, we use human-written captions from COCO [22] and VSR [23] datasets as our source captions. The LLM processes these captions to suggest edits targeting specific objects or attributes. For example, given the *source caption* "A dog in the middle of the couch", the LLM generates the *edit instruction* "change dog to doll" which contains both the *source phrase* ("dog") and the *edited phrase* ("doll"). The LLM also generates the *edited caption* "A doll in the middle of the couch". We generate five plausible (based on the criteria outlined for LLM prompting) edit instructions and edited captions per source caption. To ensure the edited captions are minimally changed w.r.t the source caption and contain visually plausible changes, we prompt the LLM again for filtering, removing $40\%$ of the total LLM outputs that do not meet those criteria (see appendix A.2.2 for details on the criteria). For **Counting and Spatial Relation** edits, we generate the source captions synthetically due to the absence of a suitable human-written captions dataset containing descriptions of counts and spatial relations of objects. We prompt the LLM to create captions and outline object layouts and bounding boxes. For instance, the LLM might generate a *caption* like "A plate of cookies on a table and a cup of coffee to its right," with the corresponding *bounding boxes*: {"plate of cookies": $[200, 300, 120, 80]$; "cup of coffee": $[320, 300, 60, 80]$}. The LLM generates a large pool of such synthetic captions. The edit instructions and the corresponding edited captions are generated using a rule-based method aimed at swapping the object positions for

spatial relation edits (e.g. *edited caption*: "A cup of coffee on a table and a plate of cookies to its right", *swapped bounding boxes*: {"1st crop": [200, 300, 120, 80]; "2nd crop": [320, 300, 60, 80]) or adjusting the object counts for counting edits (e.g. *edited caption*: "A cup of coffee on a table"), *removed bounding boxes*: {[200, 300, 120, 80]}, in this example removing the plate of cookies from the image.

**Diffusion-guided Image Synthesis** We modify images according to the edit instructions generated by the LLM in the previous step. For **Object and Attribute** edits, we first mask the object to be edited in the source image using the Grounding-DINO model [25]. We obtain the source images from the COCO dataset. The object to be edited is specified in the source phrase of the edit instruction (e.g., "a dog" in the edit instruction "change dog to doll"). We then apply the SDXL inpainting model [30], using the *input image, masked region, and edited phrase* (obtained from the edit instruction, e.g., "a doll" in the edit instruction "change dog to doll") to alter the masked image region to match the desired outcome, e.g., changing "a dog" to "a doll." For **Counting and Spatial Relation** edits, we create a synthetically generated source image dataset based on LLM-suggested layouts from the previous step, using the LLM-grounded Diffusion (LMD) model [21] for image synthesis. To create an edited image, for the spatial relation edits, we first reposition the source image's bounding boxes using a rule-based method. We then obtain image crops from the source image corresponding to the objects which we need to reposition w.r.t each other. Lastly, we use the GLIGEN layout-diffusion model [20] to smoothly insert the obtained crops into the source image at the repositioned bounding box locations. For counting edits, we obtain the edited image by always removing one or multiple objects from the source image. The object to be removed is specified by masking and we use the Lama model [34] to carry out the object removal. We employ layout-based diffusion models [21, 20] instead of using end-to-end diffusion models like Stable diffusion [30] as the layout-based model facilitates precise control over the object positions and counts and thus ensures the changes are faithful to the edit instruction as well as minimal. Unfortunately, end-to-end models such as Stable Diffusion are not good at precisely editing object positions and counts.

**Stage 2: Automatic Filtering** To ensure consistency of synthesized hard-negative images, we use a VQA-based filtering system, which is more effective than object detection (see Stage 2 in Figure 2). Questions generated by an LLM [10] based on the edit instruction and caption (following TIFA [9]) verify that edits align with the caption and that the positive caption no longer applies to the negative image. We use LLaVa 7B [24] to answer these questions, with region-specific questions for object/attribute edits and global questions for background consistency. This process removes 75% of synthesized images, ensuring dataset quality.

**Stage 3: Human Verification** To ensure high-quality of the benchmark, on top of automated filtering, we conduct human verification using the Amazon Mechanical Turk platform. The images and captions undergo four steps of verification, requiring agreement from at least four out of five annotators at each step to pass the human verification. The steps are: **1) Naturalness and Image-Text Matching Verification**: Annotators assess if (a) the image looks natural, (b) the caption is sensical, and (c) the image matches the caption. Only 26% of synthetic images pass this step, mainly due to the criterion (a), where counting and spatial relation images often look unnatural. See Appendix Table 7 for detailed acceptance rates. **2) Visual Edit Verification**: Annotators assess if the images faithfully reflect the specified minimal edits without additional changes, with an acceptance rate of 80%. **3) Edit Instruction Verification**: Annotators assess if the LLM-generated edit instructions are minimal, i.e., the suggested edit modifies only one aspect of the sentence (out of object, attribute, counting, or spatial relation), with an 84% acceptance rate. **4) Textual Edit Verification**: Annotators assess if the edited sentence faithfully reflects the specified minimal edits without additional changes, with a 95% acceptance rate. Annotators also verify the LLM's categorization of the types of edits (object, attribute, counting, or spatial relation). These steps ensure precise, minimal changes in images and captions, delivering a high-quality benchmark for fine-grained visual understanding. See Appendix A.4.1 for annotator instructions.

## 4 Training and Benchmark sets

In our study, we create training and benchmark sets to improve and assess fine-grained understanding in VLMs. The training data is generated through a scalable pipeline with automatic filtering, while the benchmark data undergoes additional rigorous human verification to ensure high quality (as explained above). For object and attribute edit types, that make use of natural images, the training data is sourced from VSR (images sourced from COCO) and the COCO 2017 training split

(118K images), while the benchmark data is sourced from the COCO 2017 validation split (5K images). This ensures benchmark images are unseen during training, maintaining evaluation reliability by community standards. The Training dataset has 64,392 samples (`37,017 objects, 10,352 attributes, 10,050 counting, 6,973 relations`), while the VisMin benchmark has 2,084 samples (`579 objects, 294 attributes, 589 counting, 622 relations`).

We aimed for a balanced benchmark across categories. However, the number of attribute samples in VisMin is relatively low because the LLM suggested attribute edits for only 2000 samples in the COCO 5K validation set. Moreover, most of these suggested edits were color edits. So we further downsampled the color edit instances to balance the distribution of the types of attribute edits. Figure 3 shows subcategories of the changes in VisMin. For detailed training set subcategories, see Appendix15. For qualitative samples, refer to Appendix 13 and 14.

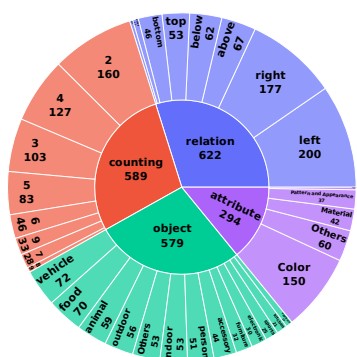

Figure 3: VisMin categories and subcategories.

In table 1, we compare VisMin with related benchmarks. **Visual Minimal HN**: This criterion evaluates if the visual hard negatives contain minimal changes. In Winoground and MMVP, hard negatives differ across multiple aspects (object, attribute, background, etc.). In contrast, VisMin's hard negatives vary only in one aspect, keeping others unchanged as much as possible. This minimal-change property is also present in What'sUp, EQBEN, SPEC, and in a subset of images from ImageCoDe and CounterCurate. **Visual Complexity**: This criterion assesses the complexity of visual scenes. ImageCoDe and EQBEN mainly feature images from limited video domains and graphic engines, while What'sUp uses simple household and tabletop images. SPEC generates simplistic scenes using diffusion models. In contrast, Winoground uses expert-curated Getty Images, and MMVP uses ImageNet and LAIONAesthetics. VisMin and CounterCurate (concurrent work) stand out by using diverse, complex everyday scenes from COCO [22] and Flicker30K Entities [29], featuring common objects in natural contexts.

**Textual Complexity**: Benchmarks like ImageCoDe, Winoground, and MMVP use free-form human-written captions. In contrast, What'sUp (focused on spatial changes) and SPEC (focused on controlled changes) use template-based captions, which often lack diversity.

Table 1: Comparison of benchmarks offering visual hard negatives (HN) across: 👁 minimal HN, 🖼 visual complexity, ✏ textual complexity, 🧍‍✔human-approved captions (❗) and images (🎨), and 📏 size. ✓: criterion holds for a subset of the benchmark.

| Benchmark | 👁 | 🖼 | ✏ | 🧍❗ | 🧍🎨 | 📏 |
|---|---|---|---|---|---|---|
| ImageCoDe [14] | ✓ | Limited video domains, Open Images | Free-form (Human) | ✓ | ✓ | 2,306 |
| What'sUp (A/B) [11] | ✓ | Household, Tabletop | Template | ✓ | ✓ | 1,232 |
| Winoground [36] | ✗ | Expert curated using Getty Images API | Free-form (Human) | ✓ | ✓ | 400 |
| EQBEN [38] | ✓ | Limited video domains, Graphic engine, Synthetic-diffusion | Free-form (Human), Template | ✗ | ✓ | 250,612 |
| SPEC [28] | ✓ | Synthetic-diffusion (limited objects) | Template | ✓ | ✗ | 3,000 |
| MMVP [37] | ✗ | ImageNet, LAIONAesthetics | Free-form (Human) | ✓ | ✓ | 150 |
| CounterCurate [45] | ✓ | Flicker30K Entities, Synthetic-diffusion | Free-form(Human, LLM), Template | ✓ | ✓ | 45,400 |
| VisMin (Ours) | ✓ | COCO, Synthetic-diffusion | Free-form (Human, LLM) | ✓ | ✓ | 2,084 |

EQBEN and CounterCurate mix free-form (human or LLM-generated) and template-based captions. VisMin combines human-written and LLM-generated free-form captions, , yielding sufficient textual complexity to the benchmark. **Human Verification**: For benchmarks using synthetic images like EQBEN, SPEC, CounterCurate, and VisMin, human evaluation is essential to ensure images look natural. It's also crucial for benchmarks with automatically generated hard negative captions, as these may be nonsensical unless well-defined templates like What'sUp are used. Nonsensical captions make it easier for VLMs to identify them as incorrect [8]. VisMin is notably the only benchmark with full human verification, ensuring both captions and images are high quality. CounterCurate also conducts human verification but only checks for image-caption consistency (on 300 examples), without verifying image naturalness or caption sensibility. **Size**: This criterion assesses the dataset's size. VisMin excels by combining **controlled minimal changes with complex, natural scenes and captions**, providing an optimal balance for robust evaluation.

## 5 Benchmarking VLMs on VisMin Benchmark

**Setup** We have comprehensively benchmarked existing *state-of-the-art* VLMs on VisMin, encompassing both foundational VLMs–such as CLIP [31], SigLip [44], BLIP [18], and Coca [40] and generative MLLMs including Llava [24], Idefics2 [15] and InternVL1.5 [2]. Additionally, closed-source MLLMs such as GPT4-o [1] and Gemini1.0 Pro [35] are also evaluated.

For foundational models like CLIP, we conducted an image-text matching task using cosine similarity, following [36]. The tasks involved two settings: choosing the correct image from two captions and selecting the correct caption from two images. In VisMin examples (see Fig. 1) with pairs $\{(I_1, C_1), (I_2, C_2)\}$, the **text score** is 1 if $(s(C_0, I_0) > s(C_1, I_0)) \wedge (s(C_1, I_1) > s(C_0, I_1))$, and the **image score** is 1 if $(s(C_0, I_0) > s(C_0, I_1)) \wedge (s(C_1, I_1) > s(C_1, I_0))$; the **group score** is 1 when both scores are 1. For MLLMs, we adapted these tasks to a visual question answering format with binary questions about the matching relationship between images and captions $\{(I_1, C_1), (I_2, C_2)\}$. To calculate the **text score**, we presented the model with one image and two captions, using the prompt *"Does this image depict: {$C_1$ or $C_2$}?"*.[3] To calculate the **image score**, we presented the model with two images and one caption, using the prompt *"Which image better aligns with the description: '{C}'? The first or the second image?"*. The score is 1 if the predicted answer matches the ground truth. Once both scores are obtained, the **group score** is 1 if both individual scores are 1.

Table 2: Performance of foundational variants and MLLMs across categories on the VisMin Dataset. Columns 'I,' 'T,' and 'G' denote Image, Text, and Group scores from Winoground [36]. AVG denotes the average across columns. The best results are highlighted in **bold**.

| | Object | | | Attribute | | | S. Relation | | | Count | | | AVG |
|---|---|---|---|---|---|---|---|---|---|---|---|---|---|
| | T | I | G | T | I | G | T | I | G | T | I | G | |
| Random Chance | 25 | 25 | 16.67 | 25 | 25 | 16.67 | 25 | 25 | 16.67 | 25 | 25 | 16.67 | 22.22 |
| MTurk Human | 86.87 | 95.50 | 83.07 | 82.31 | 91.15 | 76.87 | 81.67 | 92.76 | 76.20 | 88.96 | 96.77 | 86.41 | 86.54 |
| CLIP (ViT-B/32) [31] | 79.62 | 77.89 | 67.53 | 72.11 | 65.99 | 55.1 | 8.2 | 4.34 | 0.48 | 31.24 | 20.71 | 10.53 | 41.15 |
| CLIP (ViT-B/16) | 86.53 | 79.1 | 71.68 | 70.75 | 65.31 | 52.38 | 8.84 | 3.22 | 0.8 | 34.3 | 22.58 | 13.58 | 42.42 |
| CLIP (ViT-L/14) | 87.56 | 83.59 | 78.07 | 74.49 | 69.73 | 57.82 | 9.16 | 4.66 | 1.45 | 37.01 | 30.56 | 18.17 | 46.42 |
| NegCLIP [42] | 87.74 | 87.05 | 80.66 | 81.63 | 80.27 | 71.77 | 10.13 | 4.66 | 1.13 | 55.01 | 57.72 | 42.28 | 48.96 |
| SigLip (ViT-B/16)[44] | 90.5 | 88.95 | 83.25 | 86.05 | 79.25 | 73.13 | 11.58 | 6.43 | 1.77 | 60.95 | 47.03 | 38.37 | 55.61 |
| SigLip (ViT-L/16) | 93.44 | 88.43 | 84.46 | 84.35 | 78.23 | 68.37 | 10.29 | 4.82 | 1.29 | 61.8 | 57.05 | **44.14** | **56.39** |
| Flava [32] | 81.69 | 75.12 | 66.66 | 74.49 | 60.54 | 52.04 | 6.75 | 5.79 | 0.96 | 35.99 | 27.5 | 15.45 | 41.91 |
| BLIP [18] | 92.4 | 92.57 | **87.05** | 88.44 | 86.73 | **78.57** | 11.25 | 4.98 | **2.09** | 52.97 | 46.01 | 33.28 | 56.36 |
| Coca [40] | 84.97 | 81.52 | 73.58 | 78.57 | 66.33 | 57.82 | 11.25 | 5.95 | 1.77 | 60.1 | 35.82 | 28.52 | 48.85 |
| LlaVa1.6 (7B) [24] | 93.0 | 32.8 | 32.2 | 92.2 | 34.4 | 33.3 | 91.8 | 7.8 | 7.4 | 73.6 | 25.0 | 20.2 | 38.28 |
| Idefics2 (8B) [15] | 95.4 | 69.4 | 67.6 | 89.1 | 71.4 | 67.0 | 18.6 | 18.8 | 4.8 | 72.2 | 50.6 | **47.0** | **55.99** |
| CogVLM (7B) [39] | 94.64 | 89.63 | **87.56** | 89.11 | 88.43 | **81.63** | 21.70 | 11.09 | 2.25 | 58.40 | 6.45 | 3.56 | 52.87 |
| InternVL1.5 (25B) [2] | 94.65 | 40.24 | 39.72 | 91.16 | 42.86 | 41.16 | 74.28 | 14.79 | **11.74** | 73.51 | 31.58 | 27.5 | 48.60 |
| Gemini1.0 Pro 🔒 | 94.99 | 79.97 | 78.76 | 91.84 | 74.83 | 72.45 | 52.57 | 15.43 | 9.81 | 67.74 | 44.14 | 37.52 | 49.63 |
| GPT4-o 🔒 | 95.51 | 96.2 | **93.44** | 92.18 | 90.48 | **87.07** | 89.07 | 50.48 | **46.78** | 77.42 | 78.27 | **68.42** | **73.93** |

**Results**  Insights from Table 2 highlight key capabilities and limitations of current models. Text scores generally surpass image scores, especially in MLLMs, where text scores are often two to three times higher. In contrast, foundational VLMs show a modest discrepancy between image and text scores. We hypothesize that for MLLMs, the image score is lower compared to the text score because they lack training with multiple images, and simple vertical concatenation does not provide sufficient visual signals, leading to suboptimal alignment with captions. Notably, Idefics2, which supports multi-image processing, performs similarly on text and image scores, underscoring the importance of multi-image data during pretraining. Foundational VLMs' higher text scores suggest that distinguishing between captions is easier than between images, highlighting the need for our visual minimal change benchmark. Interestingly, foundational VLMs generally outperform MLLMs due to the latter's lower image scores.

All models perform well on Object and Attribute splits, indicating that understanding semantic changes correlates strongly with recognition capabilities. Models excelling in image classification tend to perform better, reflecting a foundational understanding that does not require advanced reasoning. For instance, Idefics2, using the SigLip (ViT-L/16) vision encoder, performs worse with strong LLMs compared to its foundational VLM counterpart, likely due to limited multi-image understanding in MLLMs. Notably, CogVLM shows superior performance over other MLLMs in understanding object and attribute changes. This advantage likely stems from its integration of grounding tasks and high-quality human-annotated datasets like Flickr30K Entities [29], RefCOCO [12], and VisualGenome [13] in its pretraining. On the other hand, the Spatial Relation split relies heavily on reasoning capabilities, with MLLMs outperforming foundational models. This suggests that LLMs can parse object relationships through reasoning. However, existing VLMs struggle with spatial relations, often scoring below random chance, indicating potential biases in models and highlighting an area for further research.

---

[3]For single-image models, such as Llava, we combine the two images vertically with a clear border.

We document human baseline performance on our benchmark via Amazon Mechanical Turk (see Appendix A.4.2). Humans generally outperform models on image scores, except in the attribute category where GPT4-o excels. Models typically surpass humans in text scores, especially with attributes and objects. However, in spatial relations and counting, humans significantly outperform models in group scores, highlighting areas for model improvement and the robustness of human scene comprehension.

# 6 Enhancing fine-grained understanding in VLMs

We use a synthetic minimal-change dataset to enhance fine-grained understanding through additional fine-tuning of VLMs. Training with pairs of images and captions with minimal differences provides a richer training signal, improving model performance in fine-grained understanding tasks. We demonstrate improvements on top of both foundational VLMs and MLLMs by conducting extensive evaluations across various benchmarks: (1) **Single image benchmarks** test the alignment between single images and multiple captions: VSR [23], CountBench [26], VALSE [27], SPEC [28], and Sugarcrepe [8]. (2) **Multiple image benchmarks** test the alignment between multiple images and captions: ImageCode [14], MMVP [37], Whatsup [11], Winoground [36], EQBEN [38], and our VisMin benchmark.

## 6.1 Fine-tuning Foundational VLMs

**Enhancement with Minimal-Change Data** Our approach uses a synthetic minimal-change dataset to improve visual representation without altering the training methodology. We construct training batches with both source and edited image-text pairs: In the original CLIP training, a mini-batch is $\mathcal{B} = \{(C_1, I_1), (C_2, I_2), \ldots, (C_n, I_n)\}$, with pairs randomly sampled from the dataset as random negatives. With minimal-change data, we add edited image-text pairs as hard negatives, resulting in $\mathcal{B} = \{(C_1, I_1), (C'_1, I'_1), (C_2, I_2), (C'_2, I'_2), \ldots\}$, where $(C'_n, I'_n)$ is the edited pair of $(C_n, I_n)$. We use a total batch size of 128 with 4 A100 GPUs and retain other training protocols and hyperparameters as default from OpenCLIP [3], including a learning rate of 1e-05, weight decay of 0.2, Adam $\beta_1$ of 0.9, $\beta_2$ of 0.98, an eps of 1e-06, and a cosine scheduler. The training runs for 5 epochs, and we select checkpoints based on a separate VisMin validation set.

We fine-tuned pre-trained CLIP on our minimal-change data, calling it VisMin-CLIP. For comparison, we implemented three models using the same pre-trained CLIP: NegCLIP [42], CounterCurate-CLIP [45], and SPEC-CLIP [28]. In NegCLIP, we fine-tuned CLIP with automatically generated hard-negative captions and nearest-neighbor images paired with human-written captions. For CounterCurate-CLIP, we used hard-negative data (attribute, position, counting) but trained one model on all three types, unlike the original approach of training separate models. SPEC-CLIP was fine-tuned on six combined category-specific splits (size, spatial, existence, count). Hard-negatives were included in all batch constructions, with excess negatives rolled into the next batch if needed. All models used ViT-L/14 as the backbone and the original CLIP loss, initialized from OpenAI checkpoints. Best checkpoints were selected from validation sets for NegCLIP and CounterCurate-CLIP, and from average benchmark performance for SPEC-CLIP. This controlled comparison evaluates the impact of different hard-negative data approaches on improving CLIP's fine-grained understanding.

Table 3: Performance of fine-tuned CLIP and Idefics2 across categories on the VisMin Dataset. The [†] symbol indicates the reproduced model checkpoints based on their respective training data.

| | Object | | | Attribute | | | S. Relation | | | Count | | | AVG |
|---|---|---|---|---|---|---|---|---|---|---|---|---|---|
| | T | I | G | T | I | G | T | I | G | T | I | G | |
| CLIP(ViT-L/14) | 87.56 | 83.59 | 78.07 | 74.49 | 69.73 | 57.82 | 9.16 | 4.66 | 1.45 | 37.01 | 30.56 | 18.17 | 46.02 |
| NegCLIP[†] | 87.74 | 87.05 | 80.66 | 81.63 | 80.27 | 71.77 | 10.13 | 4.66 | 1.13 | 55.01 | 57.72 | 42.28 | 55.00 |
| CounterCurate-CLIP[†] | 89.81 | 91.02 | 84.46 | 82.99 | 80.27 | 72.79 | 20.1 | 11.41 | **7.4** | 49.24 | 45.16 | 31.92 | 55.54 |
| SPEC-CLIP[†] | 86.53 | 86.01 | 78.58 | 78.57 | 71.77 | 63.95 | 9.16 | 5.31 | 1.13 | 45.5 | 47.71 | 32.43 | 50.55 |
| VisMin-CLIP | 91.54 | 91.19 | **86.36** | 85.03 | 83.67 | **75.85** | 11.9 | 3.38 | 1.29 | 82.34 | 79.97 | **72.33** | 63.74 |
| Idefics2 | 95.4 | 69.4 | 67.6 | 89.1 | 71.4 | 67.0 | 18.6 | 18.8 | 4.8 | 72.2 | 50.6 | 47.0 | 55.99 |
| VisMin-Idefics2 | 96.5 | 95.7 | **93.3** | 91.2 | 91.8 | **86.7** | 83.0 | 76.0 | **69.3** | 85.4 | 87.8 | **80.5** | **86.43** |

**Results** We evaluated these models on the VisMin benchmark (results in Table 3). Fine-tuning with minimal-change data significantly improves CLIP's performance on the Object, Attribute, and Count categories, demonstrating the usefulness of our minimal-change data in enhancing the fine-grained

Table 4: Evaluation on other single and mult-image **visual** fine-grained understanding benchmarks. All models adopt **ViT-L-14** as the vision encoder. **CB** refers CountBench, **SG** refers to Sugarcrepe, **IC** refers to Imagecode. I2T and T2I indicate standard standard image-to-text and text-to-image retrieval metrics. Best-performing models in the CLIP-family are highlighted in blue, and best-performing MLLM models are highlighted in green.

| | #Samples | SINGLE-IMAGE | | | | MULTI-IMAGE | | | | | | | | | | | | | |
| --- | --- | --- | --- | --- | --- | --- | --- | --- | --- | --- | --- | --- | --- | --- | --- | --- | --- | --- | --- |
| | | VSR | CB | VALSE | SG | Whatsup | SPEC | | IC | MMVP | Winoground | | | EQBEN | | | VisMin | | |
| | | | | | | | I2T | T2I | | | T | I | G | T | I | G | T | I | G |
| CLIP (ViT-L/14) | - | 58.33 | 33.65 | 69.1 | 73.0 | 37.7 | 32.85 | 30.86 | 61.47 | 19.26 | 27.5 | 11.0 | 8.5 | 35.71 | 33.57 | 21.43 | 52.05 | 47.13 | 38.88 |
| NegCLIP† | 118K | 56.56 | 40.0 | 75.41 | 85.73 | 41.2 | 37.73 | 35.45 | 67.33 | 29.63 | 25.25 | 12.0 | 7.0 | 42.86 | 40.0 | 30.0 | 58.63 | 57.42 | 48.96 |
| CounterCurate-CLIP† | 241k | 56.74 | 30.79 | 68.47 | 83.66 | 44.29 | 37.99 | 35.24 | 65.81 | 25.19 | 28.0 | 13.25 | 9.0 | 45.0 | 33.57 | 28.57 | 60.88 | 56.68 | 49.1 |
| SPEC-CLIP† | 637k | 64.54 | 32.06 | 68.75 | 79.34 | 43.35 | 87.04 | 88.08 | 66.25 | 30.37 | 22.5 | 7.75 | 4.75 | 41.43 | 40.0 | 30.71 | 54.94 | 52.7 | 44.02 |
| VisMin-CLIP | 65k | 58.69 | 49.84 | 72.24 | 81.44 | 43.99 | 44.28 | 39.98 | 66.81 | 32.59 | 32.75 | 14.75 | 9.75 | 54.29 | 40.71 | 33.57 | 67.7 | 64.55 | 58.96 |
| Idefics2 | - | 77.3 | 91.11 | 88.91 | 90.45 | 68.04 | 74.37 | 60.5 | 64.4 | 48.15 | 47.25 | 33.75 | 22.5 | 62.88 | 33.33 | 25.76 | 68.83 | 52.56 | 46.6 |
| Vismin-Idefics2 | - | 80.32 | 93.97 | 86.08 | 91.14 | 74.42 | 76.2 | 76.58 | 70.7 | 48.89 | 47.0 | 35.75 | 22.5 | 64.39 | 54.55 | 49.24 | 89.01 | 87.83 | 82.44 |

understanding of foundational VLMs such as CLIP. VisMin-CLIP consistently outperforms NegCLIP, CounterCurate-CLIP and SPEC-CLIP across all categories except spatial relations. This suggests that the visual minimal-change data is more helpful in improving the fine-grained understanding capabilities of the CLIP model compared to the nearest neighbor images in NegCLIP and not fully minimally-changed CounterCurate and SPEC data.

We further conduct a zero-shot evaluation of the fine-tuned CLIP models on other fine-grained understanding benchmarks (beyond our VisMin benchmark) to test their generalization capabilities (see Table 4). VisMin-CLIP performs best in 11 out of 18 tasks, while NegCLIP, CounterCurate-CLIP, and SPEC-CLIP lead in 3, 1, and 3 tasks, respectively. All models outperform the pre-trained CLIP model across benchmarks. For counting and spatial reasoning tasks, VisMin training data shows significant improvements. On CountBench, we observed $9\%$, $19\%$, and $17\%$ gains over NegCLIP, CounterCurate-CLIP, and SPEC-CLIP, respectively. Similarly, on spatial reasoning benchmarks (SPEC, Whatsup, VSR), VisMin-CLIP showed an average $7.79\%$ improvement over NegCLIP and $5.21\%$ over CounterCurate-CLIP. While SPEC-CLIP performed best on the in-distribution SPEC benchmark, VisMin-CLIP still outperformed other models, indicating its effectiveness in improving fine-grained understanding. For VALSE and SugarCrepe, NegCLIP performed best, likely due to similarities in the textual hard-negative generation process used in those benchmarks and NegCLIP's fine-tuning data.

Furthermore, our minimal-change data significantly outperforms others in multi-image understanding benchmarks. Fine-tuning on this data enhances the model's ability to distinguish between similar images, showing improvements on challenging benchmarks like Winoground, MMVP, and EQBEN, which test compositional reasoning and fine-grained understanding. VisMin-CLIP improved Text scores by $6\%$ on Winoground and $18\%$ on EQBEN over baseline CLIP, demonstrating effective alignment of visual and textual feature spaces. VisMin-CLIP also outperforms other models on most tasks and achieves comparable results on remaining benchmarks (except SPEC), despite using fewer samples (e.g., $65K$ in VisMin vs. $637K$ in SPEC).

Figure 4: VisMin fine-tuned models show greater improvements with larger models. The circle radius reflects the number of model parameters.

Figure 5: (left) Recall results with ViT-L/14 on COCO benchmark (right) standard benchmark results on Idefics2.

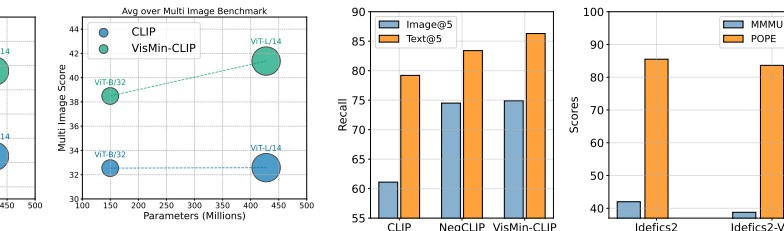

**Additional Findings** Further experiments reveal several key findings: (1) **Scalability**: As illustrated in fig. 4, we evaluated varying sizes of OpenAI's CLIP models– B/32 and L/16. Larger models demonstrated improved performance across both single-image and multi-image benchmarks after training on our synthetic data. This improvement is likely because understanding minimal changes is a complex task, demanding robust model capabilities. For instance, the smallest tested model,

ViT-B/32 (149.62M parameters), exhibited improvements of 2.37 and 3.24 in single and multiple image benchmarks, respectively, when comparing VisMin-CLIP against the baseline CLIP. When the model's capacity was expanded to ViT-L/14 (427.62M parameters), the improvements increased to 6.88 and 9.21, respectively. These results highlight the scalability and efficacy of our data in enhancing model performance. (2) **Enhanced Original Capabilities**: In addition to improvements in fine-grained understanding tasks, training on our data also enhances performance in standard retrieval tasks, as shown in fig. 5. This suggests that models achieve better alignment from training on minimal change tasks, indicating that our data is generally applicable across various cross-modal tasks.

## 6.2 Fine-tuning Multimodal Large Language Models (MLLMs)

We utilized Idefics2 [15] to improve fine-grained understanding, employing our instruction-formatted dataset. Given its proficiency in multimodal interactions and advanced multi-image processing, Idefics2 was chosen for its open-source accessibility, model size and leading zero-shot performance.

**Dataset and QLoRa Fine-tuning** Our dataset, *VisMin Instruct-IT*, includes image-text pairs created using a rule-based approach (see A.2.3 for details). We reformulated these pairs for MLLMs, where the task is to select the correct image from two options based on a given caption or choose the appropriate caption for an image from two possibilities. While the base Idefics2 model was trained with a variable number of images in a sequence, we limited it to two images to include one positive and one hard negative example from VisMin. We fine-tuned the Idefics2-8B model using the QLoRa technique [4], updating adapters in the language model and modality connector including perceiver resampler with 1 A100 80GB GPU. We used 4-bit quantization, with $r = 64$ and $\alpha = 16$ for LoRa, and a learning rate of $1e - 5$. The model was fine-tuned for one epoch with an accumulated batch size of 64.

**Results** The fine-tuned Idefics2 model shows significant improvement on VisMin (see Table 3) across all categories, comparable to GPT4-o (see Table 2), demonstrating the effectiveness of our minimal-change data in enhancing MLLM fine-grained understanding. Notably, in the Spatial Relation category, gains of 64.4%, 57.2%, and 64.5% were observed for Text, Image, and Group, respectively, unlike CLIP, where fine-tuning did not improve spatial understanding. Fine-tuning Idefics2 also transfers well to other fine-grained benchmarks, achieving over 5% overall improvement (see Table 4). To assess generalization, we evaluated its zero-shot performance on non-fine-grained tasks (MMMU [41] and POPE [19]; results in Fig. 5 (right)). The model maintained comparable performance on POPE but dropped on MMMU, likely due to the binary-choice format in our fine-tuning data. This suggests further gains could be made by combining our data with instruction-tuning datasets, though this wasn't pursued due to the GPU demands of fine-tuning an 8B model with additional data.

In addition to our main results, we provide further analysis in the Appendix A.3, including detailed evaluations on additional multimodal benchmarks, zero-shot image classification task, and video understanding tasks.

## 7    Conclusion and Limitations

We present VisMin , a benchmark for evaluating fine-grained visual understanding in VLMs such as CLIP, SigLIP, LLaVA, and Idefics2. While these models perform well in object and attribute recognition, they struggle with counting and spatial relationships. To address this, we fine-tuned CLIP and Idefics2 on our minimal-change dataset, yielding significant improvements in objects, attributes, and counting. For spatial relations, CLIP showed limited gains, while Idefics2 made notable progress. Fine-tuning also enhanced CLIP's image-text alignment, demonstrated in COCO retrieval tasks, underscoring our dataset's value as a training resource for VLMs. **Limitations:** Despite automatic filtering, the dataset contains noise, including image deformations and text-image mismatches due to diffusion model limitations. Future diffusion model advancements should improve minimal-change editing. Additionally, our model may inherit social biases from the base models (CLIP/Idefics2), as no specific mitigation measures were applied during fine-tuning. Our use of uniform prompts for evaluation may have influenced performance variably.

## Acknowledgments

We express our gratitude to Mojtaba Faramarzi and Yash Goyal for their constructive feedback throughout the different stages of the project. We also acknowledge the valuable feedback provided by Qian Yang, Kanish Jain, and Oscar Manas on the early draft. The technical support extended by the Mila IDT team in managing the computational infrastructure is greatly appreciated. Additionally, Aishwarya Agrawal received support from the Canada CIFAR AI Chair award throughout this project.

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

# A    Appendix

## A.1    Societal Impact

Our work shares the broader positive societal implications of vision-language research, particularly in improving the accuracy of vision-language models (VLMs). These advancements could benefit various applications, such as helping visually impaired individuals better interpret their surroundings, creating interactive learning tools for children, and enabling more effective interaction with in-home robots. However, like most technologies, accurate vision-language systems also present potential risks. For instance, they could be misused to extract sensitive information from public CCTV footage or inadvertently leak personal data, such as credit card information, when assisting visually impaired users. To address these concerns, robust privacy safeguards are essential to mitigate these risks.

## A.2    Prompt Templates

### A.2.1    Prompts Used for Edit Instruction Generation

We utilize the Mixtral 47B LLM to generate edit suggestions across selected categories: objects, attributes, counting, and spatial relations. The specific prompts used for each category are detailed in the Boxes below. These prompts guide the LLM in creating minimal, precise modifications to captions, ensuring the generation of high-quality hard-negative instances.

> **⊜  In-context demonstrations for objects edit suggestions.**
>
> ```
> As an AI language model, your task involves making minimal, targeted
> edits to a caption that describes an image, guiding corresponding visual
> changes to be made in the edited version of the image.
> Follow these structured steps:
> - Suggest Edit:  Identify and suggest a specific edit from the given
> caption.  Your edit should only change an object in the scene.
> *The edit must adhere to the following criteria:
> Object changes should be visually distinct and mutually exclusive.
> Do not replace an object with its synonyms or closely related
> sub-categories.  The replacement object must fit within the original
> object's region and be visually plausible within the scene.  Maintain a
> one-to-one replacement ratio; do not introduce additional objects.
> - General Criteria for Edits:  Avoid any changes related to action,
> spatial relations, counting, or the size of objects.  Edits must be
> visually and contextually plausible, ensuring the scene remains coherent
> and the edit does not introduce inconsistencies in other parts of the
> image.  Create an Edited Caption:  Draft a new caption reflecting the
> image post-edit.
> - Specify Edit Category:  Clearly state that your suggested edit falls
> under 'Object'.
> ```

> **⊜  In-context demonstrations for attributes edit suggestions.**
>
> ```
> As an AI language model, your task involves making minimal, targeted
> edits to a caption that describes an image, guiding corresponding visual
> changes in the edited version of the image.
> Follow these structured steps:
> - Suggest Edit:  Identify and suggest a specific edit from the given
> caption if it is editable.  Your edit should modify an attribute of
> existing objects in the caption.
> *The edit must adhere to the following criteria:  Change only the
> attributes of an object from one of the allowed sub-category:  color,
> pattern, shape, appearance, material, state, condition.  Replacing an
> object itself or introducing new attributes is not allowed.  Attribute
> ```

changes must be distinct, mutually exclusive, and not synonymous.  The
edited attribute should make visual and contextual sense within the
image without altering its overall composition.
- General Criteria for Edits:  Do not suggest attribute changes related
to spatial relations, counting, or the size of objects.  Edits must be
visually and contextually plausible, ensuring the scene remains coherent
and the edit does not introduce inconsistencies.  Create an Edited
Caption:  Draft a new caption reflecting the image post-edit.
- Specify Edit Category:  Clearly state that your suggested edit falls
under 'Attribute'.  If you cannot change an attribute due to a missing
attribute in the given caption, please output 'EMPTY'.

---

🛢 **In-context demonstrations for counting image description generation.**

*Task Instruction Counting*
Your goal is to suggest a textual description for creating a visual
scene within a 512x512 pixel canvas, focusing specifically on a counting
task involving objects numbered between two and nine.  Follow these
steps to ensure comprehensive output:
1.  **Input**:  Propose a scene with groups of distinct objects,
ensuring each group contains a different number of items ranging
from two to nine.  These objects should be easily countable and
distinguishable.  The objects should be spread out across the canvas;
cluttering or overlapping is prohibited.  Detail the arrangement
of these objects within the scene, considering their visibility for
accurate counting.  Do not mention any activity in the scene.
2.  **Bounding Boxes and Object Counts**:  For each group of objects,
provide bounding box data in the format 'object name', [x, y, width,
height].  This data should reflect the object's position and size within
the 512x512 image, aiding in identifying the exact number of objects
present.
3.  **Background Prompt**:  Suggest a fitting and simple background that
does not detract from the main task of counting the objects.
4.  **Negative Prompt**:  List elements that should be avoided in the
image to ensure clarity and focus on the counting task.
5.  **Category (sub-category)**:  Identify the most relevant category
and sub-category for the scene, based on the objects and their
arrangement.  Choose categories that enhance the clarity and focus of
the counting task.

---

🛢 **In-context demonstrations for relation image description generation.**

*Task Instruction Relation*
Your goal is to suggest a textual description and structured data for
creating a visual scene within a 512x512 pixel canvas.  Follow these
steps to ensure a comprehensive output:
1.  Input:  Suggest a description with pairs of distinct objects that
commonly occur together in a scene.  These objects could be swapped in
their positions if needed later on.  Avoid suggesting common background
objects in this step.  You may occasionally mention the relative size,
distance and orientation between objects if applicable.
2.  Bounding Boxes and Object Pairs:  Detail each identified object with
its bounding box in the format 'object name', [x, y, width, height],
indicating the object's position and size within the 512x512 image.  Be
specific about the spatial relationship of objects, such as whether they
are on the 'top', 'down', 'above', 'below', 'left', or 'right' of each

```
other.
3.  Background Prompt:  Based on the scene described in the input,
suggest a fitting and straightforward background.
4.  Negative Prompt:  List elements to be excluded from the image.
5.  Category (sub-category):  Identify the most relevant category and
sub-category for the scene based on the objects and their arrangement.
Limit the options for sub-categories based on the provided reference
examples.
```

### A.2.2  Prompts Used for Edit Instruction Verification

We employ the Mixtral47B LLM to verify edit suggestions in the object and attribute categories. The specific prompts used for this verification process are illustrated in the Box A.2.2. Verification is crucial because initial suggestions by the LLM can occasionally be implausible or violate the intended edit category. By thoroughly verifying the edits, we ensure the generation of accurate and high-quality hard-negative instances, which are essential for effectively testing and improving VLMs.

> **In-context demonstrations for LLM suggested edit verification for object and attribute category.**
>
> ```
> **Task Instruction Object**
> Determine the acceptability of edits to an image caption based on
> following criteria:
> - Edits must only involve object changes within a specified scene region.
> - No changes related to attributes, spatial relations, counting, or any
> unrelated alterations.
> - The new object must be visually distinct and not closely related to
> the original.  Edits should be confined to one region without needing
> adjustments elsewhere for scene consistency.
> - Adhere to a one-to-one replacement rule, introducing no additional
> objects.  Reject edits that add unnecessary specificity or make
> assumptions not clear from the original caption.
> - Reject if information is insufficient or uncertain.  Strict filtering:
> When in doubt, reject the edit.
>
>
> **Task Instruction Attribute**
> Evaluate edits to an image caption that change an object's attributes
> based on following criteria:
>
> - Reject changes involving the objects themselves, their spatial
> relations, counting, or size.
> - Acceptable modifications are distinct changes in color, pattern, shape,
> texture, material, state, or condition.  Attribute changes must be
> distinct and not synonymous.
> - Reject edits suggesting synonym replacements or adding new attributes.
> - Reject if information is insufficient or uncertain.  Strict filtering:
> When in doubt, reject the edit.
> ```

### A.2.3  VisMin Instruct-It Dataset

In the VisMin Instruction dataset, a sample consists of a source image-caption pair $(I_0, C_0)$ and its minimally edited version $(I_1, C_1)$. To construct the VisMin Instruct-IT dataset, we create four instruction samples from each VisMin sample:

- Two samples for the Image-Task (selecting the best matching image given a caption):
    1. Given $C_0$, select between $I_0$ and $I_1$
    2. Given $C_1$, select between $I_0$ and $I_1$
- Two samples for the Text-Task (selecting the best matching caption given an image):

1. Given $I_0$, select between $C_0$ and $C_1$
2. Given $I_1$, select between $C_0$ and $C_1$

Please refer to Table 5 in the rebuttal PDF for the exact set of instruction templates used. We randomly sample one template for each task type when creating the instruction samples.

In total, we create 65k instruction samples for training. Additionally, we include 16k randomly selected image-caption pairs (either $(I_0, C_0)$ or $(I_1, C_1)$) to retain the model's general image captioning ability.

Table 5: VisMin Instruct-It dataset creation templates

| Template Type | Template Examples |
|---|---|
| Image-Task | "You are given two images. Which one better aligns with the description: {caption}? The first or the second image?" 
 "Question: You are given two images. Which one better aligns with the description? {caption} Choices: First, Second." 
 "Based on the description: {caption}, please select one of the given options that best matches the image. Choices: First, Second." |
| Text-Task | "Question: Does this image depict: (A) {candidate text A}, or (B) {candidate text B}? Choices: A, B." 
 "Does this image best represent: (A) {candidate text A}, or (B) {candidate text B}?" 
 "Which of the following best matches the image: (A) {candidate text A}, or (B) {candidate text B}?" |

## A.3 Additional Results

We report additional results to demonstrate the impact of fine-tuning on our dataset across different aspects of scene understanding. We evaluated our fine-tuned models, VisMin-CLIP and VisMin-Idefics2, on various tasks designed to assess distinct elements of multimodal comprehension.

Table 6: Performance breakdown of EQBEN video splits for both CLIP fine-tuned and Idefics2 models. The table also includes results on additional multimodal benchmarks for Idefics2 family of models and ImageNet results for various CLIP models. We leave the CLIP results on the additional multimodal benchmarks and the Idefics2 results on ImageNet blank due to the task-format not being suitable for evaluation.

| | EQ-YouCook2 | | | EQ-GEBC | | | EQ-AG | | | Additional Multimodal Benchmarks | | | | ImageNet | |
|---|---|---|---|---|---|---|---|---|---|---|---|---|---|---|---|
| | T | I | G | T | I | G | T | I | G | MathVista | MMBench | TextVQA | VQAv2 | top1 | top5 |
| CLIP(ViT-L/14) | 50.00 | 65.00 | 40.00 | 15.00 | 10.00 | 0.00 | 15.00 | 25.00 | 10.00 | - | - | - | - | 75.53 | 94.59 |
| NegCLIP | 65.00 | 60.00 | 50.00 | 20.00 | 20.00 | 0.00 | 15.00 | 35.00 | 10.00 | - | - | - | - | 65.50 | 90.00 |
| VisMin-CLIP | 80.00 | 70.00 | 65.00 | 30.00 | 10.00 | 5.00 | 30.00 | 25.00 | 10.00 | - | - | - | - | 68.20 | 91.81 |
| Idefics2 | 84.62 | 53.85 | 53.85 | 25.00 | 40.00 | 15.00 | 52.63 | 47.37 | 31.58 | 53.00 | 76.18 | 73.40 | 78.82 | - | - |
| VisMin-Idefics2 | 69.23 | 84.62 | 61.54 | 35.00 | 20.00 | 15.00 | 57.89 | 36.84 | 31.58 | 47.80 | 72.74 | 71.89 | 79.75 | - | - |

**Multimodal Benchmarks**: We assessed performance of multimodal LLMs (Idefics2 and Vismin-Idefics2) on benchmarks like MathVista, MMBench, TextVQA, and VQAv2 to understand how fine-tuning on our data impacts tasks requiring both general scene understanding and specialized skills (e.g., OCR and math) (see Table 6, middle block).[4] Results show slight declines on specialized tasks (MathVista, MMBench, TextVQA) but notable improvements on VQAv2 that tests for general scene understanding, supporting our hypothesis that our dataset strengthens general scene comprehension while not losing performance much on more specialized tasks.

**Zero-Shot Image Classification**: To evaluate the robustness of learned representations of the contrastive models, we included evaluation on zero-shot image classification on ImageNet (see Table 6, last block). Both NegCLIP and VisMin-CLIP underperform compared to the base CLIP model. However, VisMin-CLIP shows a smaller performance drop compared to NegCLIP, indicating that it retains a stronger generalization capability even after fine-tuning.

**Performance on Video Understanding Tasks**: Finally, we tested if fine-tuning the models on our minimal-change dataset also improves their performance on video understanding tasks since video understanding tasks require identifying subtle frame-to-frame variations (see Fig. 6 for some examples). To test this, we evaluated the models on the EQBEN [38] benchmark, particularly the following splits: EQ-YouCook2, which focuses on cooking procedures from instructional videos; EQ-GEBC, which identifies event boundaries in kinetic videos; and EQ-AG, which captures changes between objects and their pairwise relationships during actions.

---

[4]Contrastive models with two-tower encodings are not suitable for these benchmarks, as these benchmarks require generating text outputs rather than just measuring similarity between encoded images and texts.

We would like to note that our fine-tuned models excel at discerning subtle *semantic* differences, but neighboring video frames often exhibit *low-level* rather than semantic changes. Abrupt changes between frames, like a bus turning into a car, a shirt changing color, or objects swapping positions, are rare. Additionally, our dataset does not cover action changes, which are typical in videos. Therefore, we expect finetuning on our dataset to result in limited improvements for nearby video frames.

This was confirmed in our results (see Table 6, first block), where fine-tuning CLIP and Idefics2 showed significant improvements on EQ-YouCook2 (in group scores) due to its higher presence of object-based changes, while performance remained similar on EQ-GBEC and EQ-AG, which mostly feature action shifts. Thus, our dataset is most effective for enhancing video understanding in scenarios involving subtle semantic adjustments rather than dynamic actions.

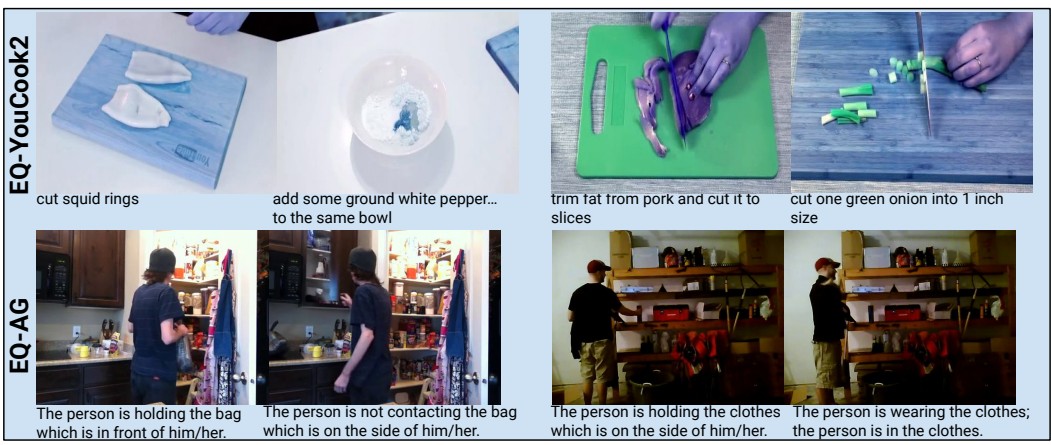

Figure 6: Examples from different video splits from the EQBEN benchmark.

## A.4    Annotation Platform and Benchmark Quality

### A.4.1    Human Verification of Benchmark

The human verification process involves four main steps. **1) Naturalness and Image-Text Matching Verification:** in this step, humans evaluate the data samples using the following three criteria: a) whether the image looks natural or not, b) whether the caption sounds sensical or not, and c) whether the image matches the caption or not, addressing limitations of automatic filtering and maintaining robustness against manipulation [8] (see Figure 7). Only 26% of synthetic images pass this step, highlighting the need for human verification. The low acceptance rate is mainly associated with the high rejection rate of criterion a, where most counting and spatial relation synthetic images do not appear natural. Please refer to Appendix Table 7 for detailed acceptance rates for each criterion. Please note that we aimed for a balanced benchmark. To address the higher rejection rates for the categories of counting and spatial relation, we began with a larger number of samples in these categories. **2) Visual Edit Verification** confirms that the edited images accurately reflect the specified edits without additional changes, ensuring visual minimal change, with an acceptance rate of 80% (see Figure 8). **3) Edit Instruction Verification** checks that LLM-generated edit instructions are minimal and targeted, altering only one aspect of the object, attribute, counting, or spatial relation in captions and images, with an acceptance rate of 84%. (see Figure 9). **4) Textual Edit Verification** ensures the edited sentence accurately reflects the specified edit instruction without additional changes and classifies the change type, with an acceptance rate of 95% (see Figure 10).

### A.4.2    Human baseline

We gather human annotations using Amazon Mechanical Turk (AMT) to determine human performance on our benchmark, replicating the exact tasks given to models as described in Section 5. Additionally, annotators can choose "none" or "both" options (please see Figure 11 and Figure 12); we deliberately included these options to accurately estimate the best model performance when tasks are difficult for humans. We collect five annotations per sample for each of the four scenarios in the image-text matching task. We compute image score, text score, and group score as described in

Table 7: Acceptance rates for all criteria across different categories.

| Criterion | Object | Attribute | Counting | S. Relation | Overall |
|---|---|---|---|---|---|
| Step 1-a: Naturalness of Image | 64% | 80% | 41% | 22% | 37% |
| Step 1-b: Sensicality of Caption | 84% | 79% | 75% | 70% | 74% |
| Step 1-c: Image-Text Matching | 83% | 89% | 52% | 65% | 65% |
| Step 2: Visual Edit Verification | 81% | 79% | 81% | 78% | 80% |
| Step 3: Edit Instruction Verification | 81% | 86% | 85% | 84% | 84% |
| Step 4: Textual Edit Verification | 95% | 94% | 94% | 95% | 95% |
| Step 4: Automatic Categorization Verification | 95% | 91% | 99% | 100% | 97% |

Section 5. If the answer "both" or "none" is chosen by three or more annotators, or if one annotator chooses "none" or "both" while the other options each receive exactly two votes, we randomly assign a 50% match to maintain consistency with the model evaluation setup.

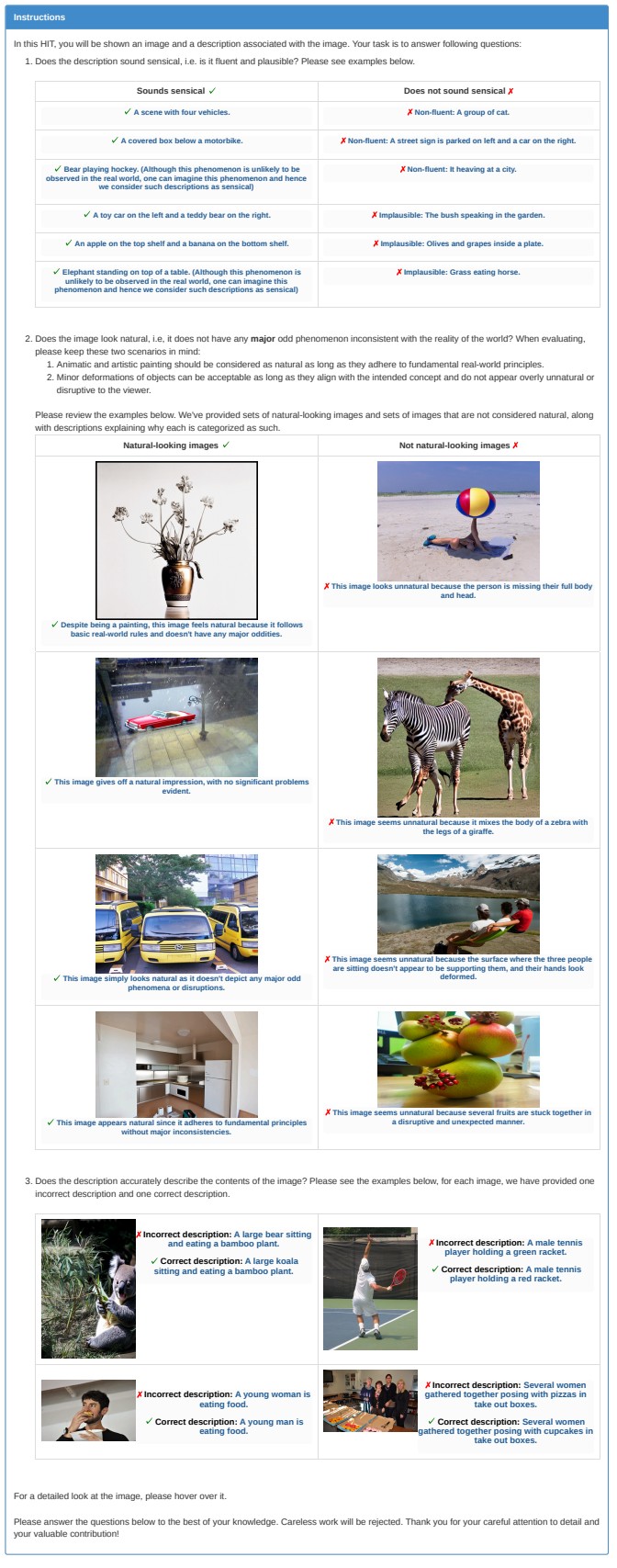

Figure 7: Instructions for naturalness and image-text matching verification.

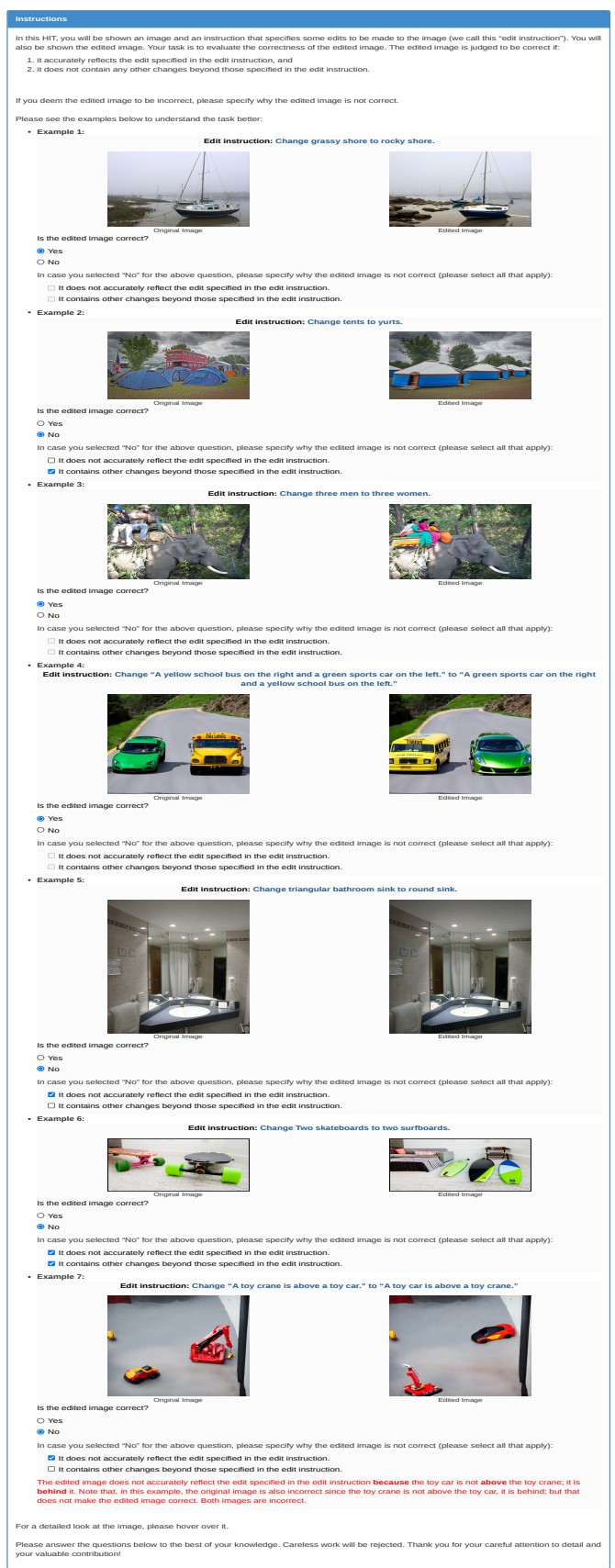

Figure 8: Instructions for visual edit verification after applying edit instructions.

**Instructions**

In this HIT, you will be shown a sentence and an instruction that specifies some edits to be made to the sentence (we call this "edit instruction"). Your task is to verify the validity of the edit instruction. The edit instruction is considered valid if all of the following conditions holds for it:

1. The suggested edit results in mutually exclusive concepts, e.g., changing dog to puppy is not a valid edit instruction since puppy is a type of dog, and hence they are not mutually exclusive. Please see the examples below to understand mutually exclusive concepts better:
   - ✅ Changing "Poodle" to "Bulldog", "Rose" to "Orchid", "woman" to "man", "car" to "truck", "cold drink" to "warm drink", "dog" to "cat", and "sofa" to "bed", are **mutually exclusive** changes.
   - ❌ Changing "dog" to "puppy", "Rose" to "flower", "woman" to "person", "car" to "vehicle", and "cold drink" to "drink" are **not mutually exclusive** changes. Please note that any replacement of words with their synonyms is also not considered mutually exclusive, e.g., changing "kids" to "children", changing "drink" to "beverage", and changing "insect" to "bug".

2. The suggested edit modifies **only** one aspect of the sentence, chosen from one of the following categories only:
   - **Object:** modifying an object (e.g., changing dog to cat).
   - **Attribute:** modifying the properties of an object, such as its color, pattern, shape, or material (e.g., changing round table to rectangular table).
   - **Counting:** modifying the count of an object (changing three dogs to two dogs).
   - **Spatial Relationship:** modifying the spatial relationship between two objects (e.g., changing "A cat to the left of the dog." to "A dog to the left of the cat.").

3. The modification type should fall within one of the **four specified categories**: object, attribute, counting, or spatial relationship. For example, changing verbs is not allowed.

If you deem the edit instruction to be invalid, please specify why the edit instruction is not valid.

Please see the examples below to understand the task better:

- **Example 1:**
  **Original sentence: A dog jumps for a frisbee over a pool.**

  **Edit instruction: Change frisbee to ball.**

  Is the edit instruction valid?
  ◉ Yes
  ○ No

  In case you selected "No" for the above question, please specify why the edit instruction is not valid (please select all that apply):
  ☐ It does not result in mutually exclusive concepts.
  ☐ It modifies more than one aspect of the sentence within the categories of object, attribute, counting, and spatial relationship.
  ☐ The modification type does not fall within any of the four specified categories: object, attribute, counting, or spatial relationship.

- **Example 2:**
  **Original sentence: A picture of a couple who got married.**

  **Edit instruction: Change couple to bride and groom.**

  Is the edit instruction valid?
  ○ Yes
  ◉ No

  In case you selected "No" for the above question, please specify why the edit instruction is not valid (please select all that apply):
  ☑ It does not result in mutually exclusive concepts.
  ☐ It modifies more than one aspect of the sentence within the categories of object, attribute, counting, and spatial relationship.
  ☐ The modification type does not fall within any of the four specified categories: object, attribute, counting, or spatial relationship.

- **Example 3:**
  **Original sentence: A group of children riding bicycles along a scenic path in the park.**

  **Edit instruction: Change bicycles to bikes.**

  Is the edit instruction valid?
  ○ Yes
  ◉ No

  In case you selected "No" for the above question, please specify why the edit instruction is not valid (please select all that apply):
  ☑ It does not result in mutually exclusive concepts.
  ☐ It modifies more than one aspect of the sentence within the categories of object, attribute, counting, and spatial relationship.
  ☐ The modification type does not fall within any of the four specified categories: object, attribute, counting, or spatial relationship.

- **Example 4:**
  **Original sentence: A picture of a white cat.**

  **Edit instruction: Change white cat to black dog.**

  Is the edit instruction valid?
  ○ Yes
  ◉ No

  In case you selected "No" for the above question, please specify why the edit instruction is not valid (please select all that apply):
  ☐ It does not result in mutually exclusive concepts.
  ☑ It modifies more than one aspect of the sentence within the categories of object, attribute, counting, and spatial relationship.
  ☐ The modification type does not fall within any of the four specified categories: object, attribute, counting, or spatial relationship.

- **Example 5:**
  **Original sentence: A purple bicycle chained to a metal tree enclosure.**

  **Edit instruction: Change purple bicycle to red bicycle.**

  Is the edit instruction valid?
  ◉ Yes
  ○ No

  In case you selected "No" for the above question, please specify why the edit instruction is not valid (please select all that apply):
  ☐ It does not result in mutually exclusive concepts.
  ☐ It modifies more than one aspect of the sentence within the categories of object, attribute, counting, and spatial relationship.
  ☐ The modification type does not fall within any of the four specified categories: object, attribute, counting, or spatial relationship.

- **Example 6:**
  **Original sentence: Three apples sitting in a bowl.**

  **Edit instruction: Change 3 apples to 2 apples.**

  Is the edit instruction valid?
  ◉ Yes
  ○ No

  In case you selected "No" for the above question, please specify why the edit instruction is not valid (please select all that apply):
  ☐ It does not result in mutually exclusive concepts.
  ☐ It modifies more than one aspect of the sentence within the categories of object, attribute, counting, and spatial relationship.
  ☐ The modification type does not fall within any of the four specified categories: object, attribute, counting, or spatial relationship.

- **Example 7:**
  **Original sentence: A flower vase on the table and a cake on the right.**

  **Edit instruction: Change "A flower vase on the table and a cake on the right." to "A cake on the table and a flower vase on the right."**

  Is the edit instruction valid?
  ◉ Yes
  ○ No

  In case you selected "No" for the above question, please specify why the edit instruction is not valid (please select all that apply):
  ☐ It does not result in mutually exclusive concepts.
  ☐ It modifies more than one aspect of the sentence within the categories of object, attribute, counting, and spatial relationship.
  ☐ The modification type does not fall within any of the four specified categories: object, attribute, counting, or spatial relationship.

  **Original sentence: The kids joyfully dancing together in the sandbox.**

  **Edit instruction: Change dancing to playing.**

  Is the edit instruction valid?
  ○ Yes
  ◉ No

  In case you selected "No" for the above question, please specify why the edit instruction is not valid (please select all that apply):
  ☐ It does not result in mutually exclusive concepts.
  ☐ It modifies more than one aspect of the sentence within the categories of object, attribute, counting, and spatial relationship.
  ☑ The modification type does not fall within any of the four specified categories: object, attribute, counting, or spatial relationship.

Please answer the questions below to the best of your knowledge. Careless work will be rejected. Thank you for your careful attention to detail and your valuable contribution!

Figure 9: Instructions for edit instruction verification.

In this HIT, you will be shown a sentence and an instruction that specifies some edits to be made to the sentence (we call this "edit instruction"). You will also be shown the edited sentence. Your task is to evaluate the correctness of the edited sentence. The edited sentence is judged to be correct if:

1. it accurately reflects the edit specified in the edit instruction, and
2. it does not contain any other changes beyond those specified in the edit instruction.

If you deem the edited sentence to be incorrect, please specify why the edited sentence is not correct.

Finally, you must select the **type** of the edit being specified in the edit instruction, from the following types:

- **Object:** modifying an object (e.g., changing dog to cat).
- **Attribute:** modifying the properties of an object, such as its color, pattern, shape, or material (e.g., changing round table to rectangular table).
- **Counting:** modifying the count of an object (changing three dogs to two dogs).
- **Spatial Relationship:** modifying the spatial relationship between two objects (e.g., changing "A cat to the left of the dog." to "A dog to the left of the cat.")

Please see the examples below to understand the task better:

- **Example 1:**
  **Original sentence: A black and white cat is sitting in front of a laptop that is on a desk.**

  **Edit instruction: Change laptop to tablet.**

  **Edited sentence: A black and white cat is sitting in front of a tablet that is on a desk.**

  Is the edited sentence correct?

  ◉ Yes
  ○ No

  In case you selected "No" for the above question, please specify why the edited sentence is not correct (please select all that apply):

  ☐ It does not accurately reflect the edit specified in the edit instruction.
  ☐ It contains other changes beyond those specified in the edit instruction.

  Finally, select the type of the edit being specified in the edit instruction:

  ◉ Object
  ○ Attribute
  ○ Counting
  ○ Spatial Relationship

- **Example 2:**
  **Original sentence: A white park bench sitting on a middle of a forest.**

  **Edit instruction: Change white park bench to red park bench.**

  **Edited sentence: A red park bench sitting on a middle of a forest.**

  Is the edited sentence correct?

  ◉ Yes
  ○ No

  In case you selected "No" for the above question, please specify why the edited sentence is not correct (please select all that apply):

  ☐ It does not accurately reflect the edit specified in the edit instruction.
  ☐ It contains other changes beyond those specified in the edit instruction.

  Finally, select the type of the edit being specified in the edit instruction:

  ○ Object
  ◉ Attribute
  ○ Counting
  ○ Spatial Relationship

- **Example 3:**
  **Original sentence: A dog sleeping on top of a red chair with two books.**

  **Edit instruction: Change two books to one book.**

  **Edited sentence: A red chair without a dog sleeping on it and without the two books nearby.**

  Is the edited sentence correct?

  ○ Yes
  ◉ No

  In case you selected "No" for the above question, please specify why the edited sentence is not correct (please select all that apply):

  ☑ It does not accurately reflect the edit specified in the edit instruction.
  ☑ It contains other changes beyond those specified in the edit instruction.

  Finally, select the type of the edit being specified in the edit instruction:

  ○ Object
  ○ Attribute
  ◉ Counting
  ○ Spatial Relationship

- **Example 4:**
  **Original sentence: A bike to the left of a door.**

  **Edit instruction: Change "A bike to the left of a door." to "A door to the left of a bike."**

  **Edited sentence: A door to the left of a bike.**

  Is the edited sentence correct?

  ◉ Yes
  ○ No

  In case you selected "No" for the above question, please specify why the edited sentence is not correct (please select all that apply):

  ☐ It does not accurately reflect the edit specified in the edit instruction.
  ☐ It contains other changes beyond those specified in the edit instruction.

  Finally, select the type of the edit being specified in the edit instruction:

  ○ Object
  ○ Attribute
  ○ Counting
  ◉ Spatial Relationship

Please answer the questions below to the best of your knowledge. Careless work will be rejected. Thank you for your careful attention to detail and your valuable contribution!

Figure 10: Instructions for textual edit verification.

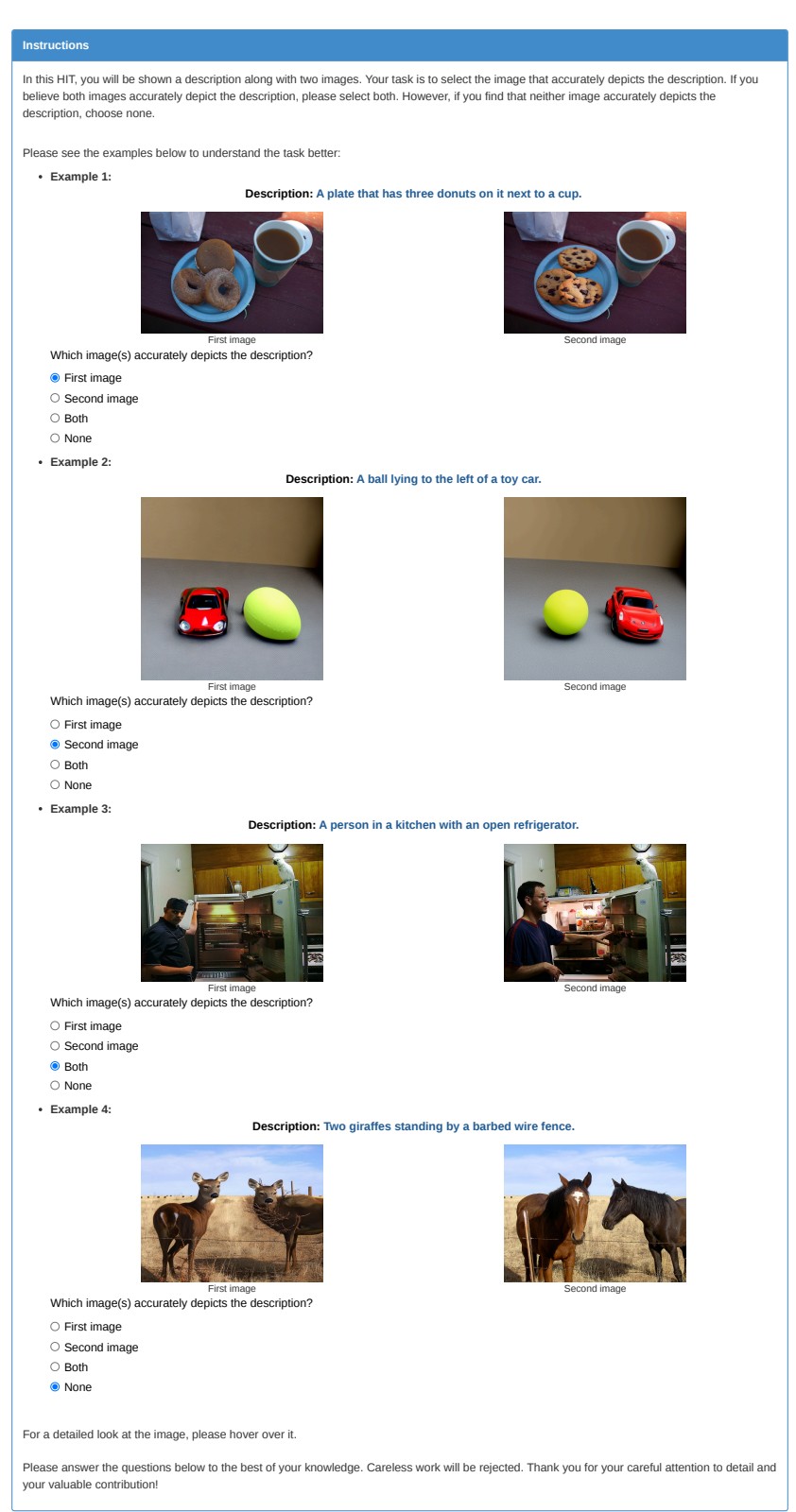

Figure 11: Instructions for choosing the best matching image.

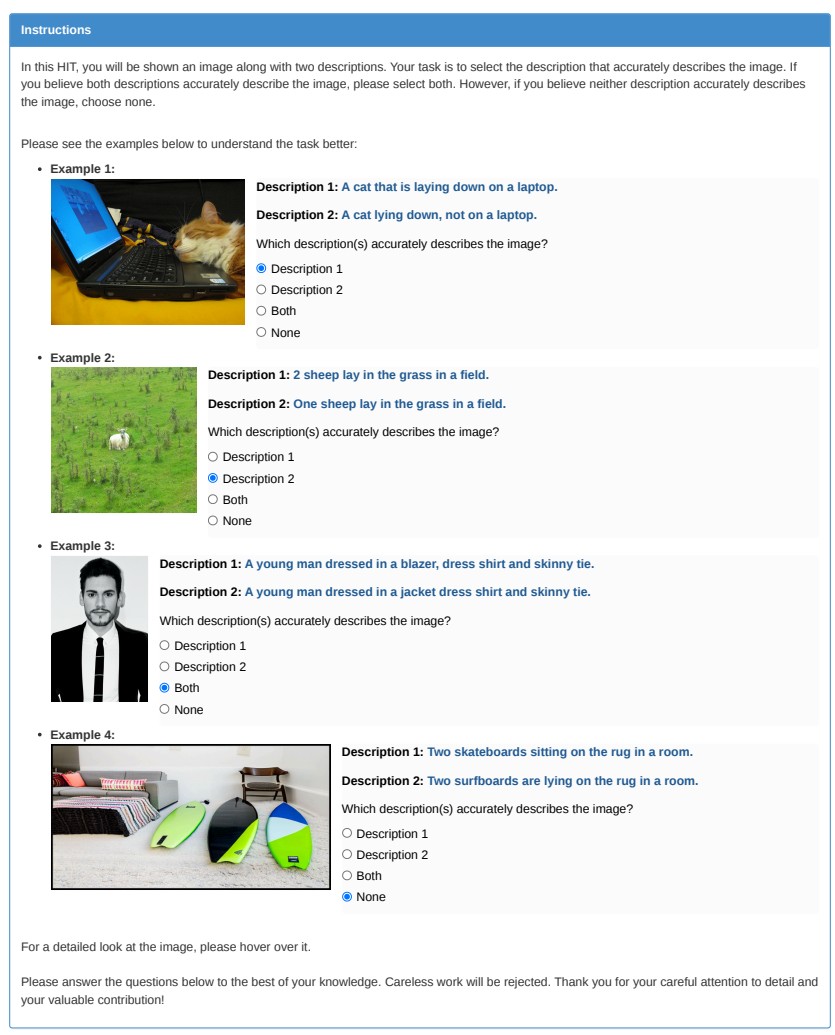

Figure 12: Instructions for choosing the best matching description.

## A.5 Random qualitative samples from training and VisMin benchmark

In this section, we present random qualitative samples from both the training set and the VisMin benchmark to illustrate the quality and diversity of data used in our experiments. Specifically, we provide examples across object, attribute, relation, and counting categories to highlight the four strategic categories represented in our dataset.

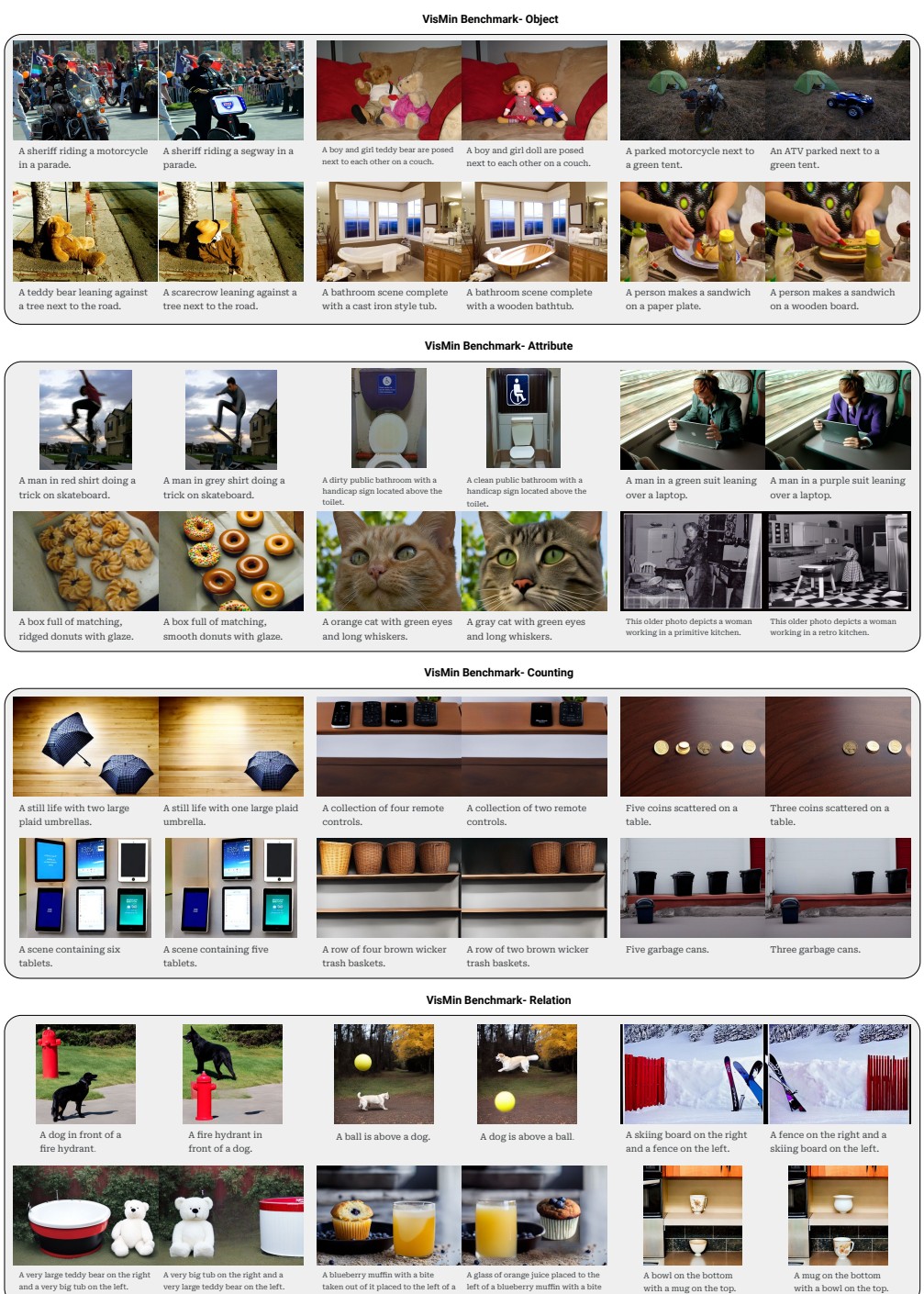

Figure 13: Random qualitative samples from VisMin.

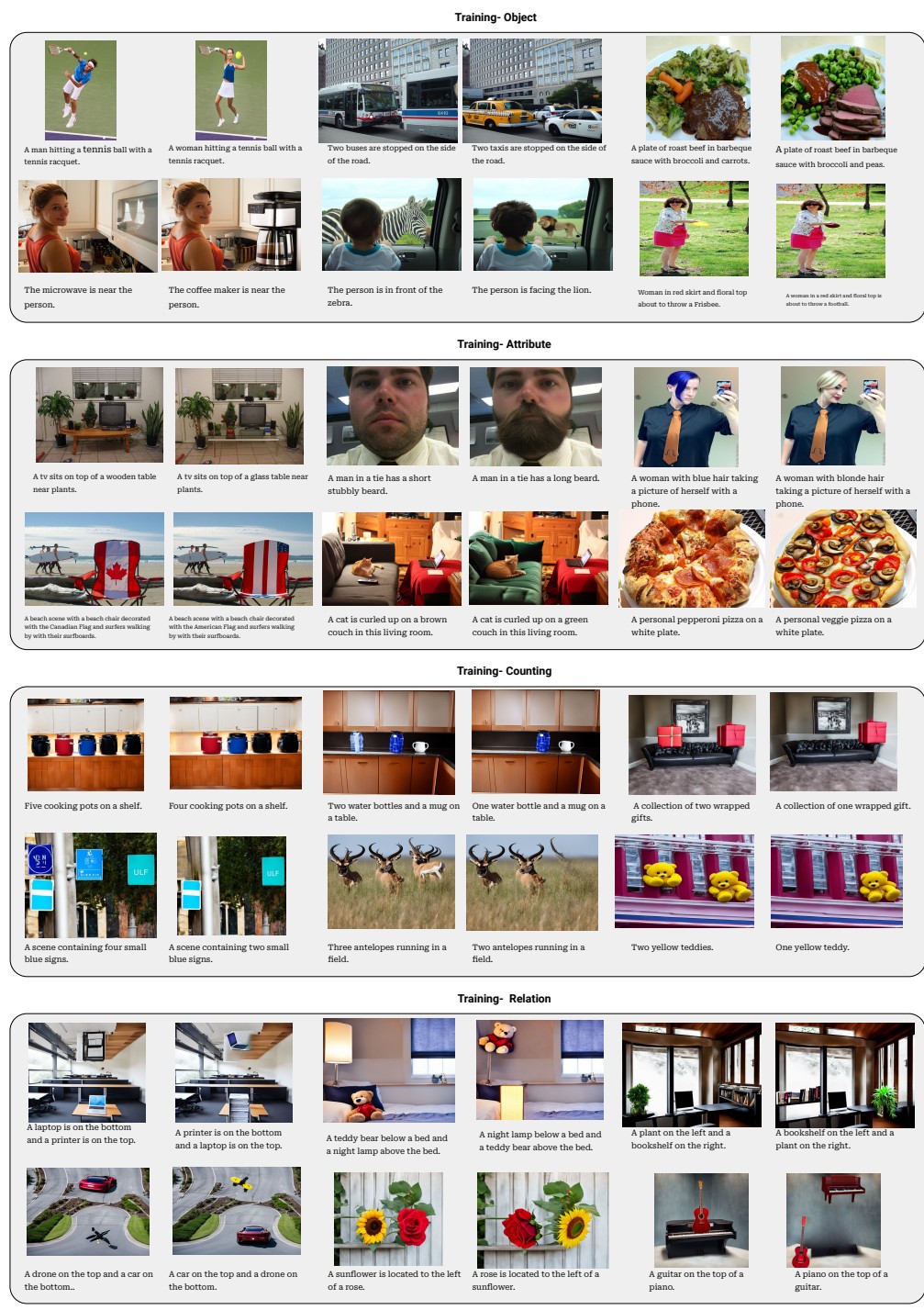

Figure 14: Random qualitative samples from the training set.

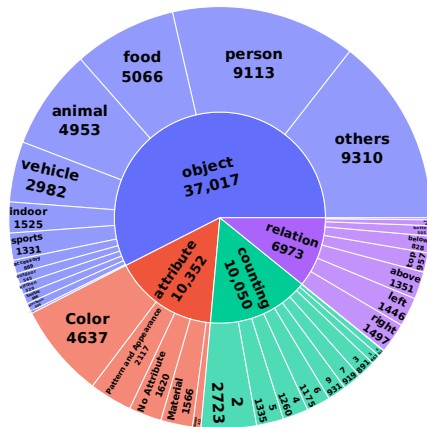

Figure 15: Categories and subcategories distribution in the training split.

