# OpenReview forum: "VisMin: Visual Minimal-Change Understanding"
_NeurIPS.cc/2024/Conference — NeurIPS 2024 poster_

### Official Review · Reviewer_dNLV · 2024-07-12

**Soundness:** 2
**Presentation:** 3
**Contribution:** 3
**Rating:** 5
**Confidence:** 5

**Summary:**

The paper introduces VisMin, a new benchmark for assessing fine-grained understanding in VLMs. VisMin evaluates the ability of models to distinguish between minimally different images given a caption, testing recognition of changes in objects, attributes, counts, and spatial relationships. The benchmark is created using an automated pipeline and human verification. Empirical results show that current VLMs have notable deficiencies in spatial relationships and counting. Fine-tuning models like CLIP and Idefics2 with a large-scale dataset generated by this pipeline leads to improvements in fine-grained understanding and image-text alignment.

**Strengths:**

1.This paper introduces VisMin, a benchmark that challenges models to detect subtle semantic differences between similar images, revealing significant shortcomings in VLMs and MLLMs.
2.This paper uses minimal-change image-text data to finetune VLM and MLLM models, improving the fine-grained understanding capabilities.
3.Table 2 illustrates that current open-source MLLMs perform poorly in fine-grained image understanding. This paper suggests using minimal changes in images to enhance fine-grained image understanding, and I believe this idea is promising.

**Weaknesses:**

1.After fine-tuning on VisMin data, MLLMs showed improved performance on fine-grained understanding benchmarks. However, their performance on POPE and MMMU decreased, which contradicts the authors' claim of improved text-image alignment in MLLMs. The authors should provide results on additional benchmarks such as TextVQA, MathVista, and MMBench.
2.The authors attribute the poor performance to the binary-choice task and limited computational resources preventing training of an 8B model. This explanation seems insufficient; given the computational constraints, the authors could reduce the model size, such as using Qwen-1.8B, to validate the method's effectiveness.
3.For VLMs, the authors should also provide some zero-shot image classification results, such as on ImageNet. I am curious to know if fine-tuning on VisMin data affects zero-shot classification performance.

**Questions:**

Please refer to Weaknesses.
There are also some typo like L76: can -> can, L169: he -> the, Duplicate citation [39], [40], Table 3 and 4: Vismin -> VisMin.

**Limitations:**

The authors have addressed the limitations.

---

> ### Author Rebuttal · Authors · 2024-08-07
>
> **“which contradicts the authors' claim of improved text-image alignment in MLLMs”**
>
> We do not claim that we improve general image-text alignment *for MLLMs*. We claim that we improve *CLIP’s* general image-text alignment and substantiate that claim by showing improvements on the standard COCO image-text retrieval.
>
> Also, we would like to note that improvement in fine-grained understanding tasks does not necessarily translate to an improvement in standard image-text understanding tasks as also observed in another recent work [1]. This is due to differences in data distributions of these tasks. For example, standard tasks have more diversity in object classes whereas fine-grained tasks have more diversity in relations, attributes etc. So the best we can hope for is that the performance on standard tasks does not drop much. And we show that this is the case for our method.
>
> [1] Bugliarello et al. Weakly-supervised learning of visual relations in multimodal pretraining. EMNLP 2023.
>
> **Results on additional benchmarks**
>
> First, we would like to clarify that the benchmarks suggested by the reviewer require certain specific skills (in addition to general image-text understanding) that our minimal change data does not cover. For instance, TextVQA requires Optical Character Recognition (OCR), MathVista requires math understanding, MMBench requires OCR, structured data understanding, and celebrity recognition. The base model (Idefics2) has been trained on datasets containing such specific skills (e.g., Idefics2’s pretraining data consists of numerous math understanding datasets [2, 3, 4]). However, our minimal change data focuses on general scene understanding. Thus, finetuning Idefics2 on our minimal change data is expected to cause catastrophic forgetting [5] of such specific skills leading to a degradation in model performance on such benchmarks. To mitigate this, one needs to continue training the base model on datasets containing such specific skills along with finetuning it on our dataset, however experimenting with that is out of the scope of this paper.
>
> As per the reviewer’s request we evaluated the models on the suggested benchmarks. In addition, we also evaluated the models on the VQA v2 benchmark which tests for general scene understanding. See Table 2 of the rebuttal PDF for the results. All results are reported on the validation split (test splits are not publicly available or require submitting to evaluation servers), except for MathVista, whose test split is publicly available. As expected, we see that finetuning on our data leads to a drop in the performance on all benchmarks except VQAv2 where we see improvements in the performance of the base Idefics2 model, thus demonstrating that our data is indeed helpful for improving general multimodal scene understanding.
>
> [2] Kazemi et al. GeomVerse. arXiv 2023.
>
> [3] Lindström et al. CLEVR-Math. arXiv 2022.
>
> [4] Lu et al. Inter-GPS. ACL 2021.
>
> [5] Kirkpatrick et al. Overcoming catastrophic forgetting in neural networks. Proceedings
> of the National Academy of Sciences 2017.
>
> **“the authors could reduce the model size, such as using Qwen-1.8B, to validate the method's effectiveness”**
>
>
> We’ve actively researched smaller models for our MLLM experiments. To the best of our knowledge, Idefics2 is the best-performing open-source model *capable of processing multiple images*. We’d like to clarify that the Qwen-1.8B suggested by the reviewer is a language model; not an MLLM. Indeed, there’s an MLLM that’s built on top of Qwen LLM called Qwen-VL [6]. However, Qwen-VL is a 7B parameter model and cannot process multiple images. We’ve again reviewed recent papers (during this rebuttal period) and we’ve not found any smaller model capable of processing multiple images. For instance, although MiniCPM [7] has models with 3B params; however none of them can process multiple images.
>
> [6] Bai et al. Qwen-vl: A frontier large vision-language model with versatile abilities. arXiv 2023.
>
> [7] Hu et al. MiniCPM: Unveiling the Potential of Small Language Models with Scalable Training Strategies. arXiv 2024.
>
> **Zero-shot image classification results**
>
> We’ve incorporated the zero-shot image classification results in the last column of table 2 of the rebuttal PDF. The results show that after fine-tuning with a hard negative dataset, both NegCLIP and VisMin-CLIP under-perform the base CLIP model on zero-shot image classification task. However, VisMin-CLIP shows a smaller performance drop compared to NegCLIP, thus demonstrating that our method learns more robust representations.
>
> Lastly, as mentioned in our previous response, the drop in the performance on standard tasks is expected when finetuned on fine-grained tasks. We agree that an ideal model should work well for both types of tasks, however, given the large improvements of our method for fine-grained tasks and also for standard COCO image-text retrieval and VQA v2 tasks, we believe that the contribution of our method is still quite positive.
>
> **“There are also some typo”**
>
> We apologize for these errors and thank the reviewer for pointing them out. We will fix them in the camera-ready version.

---

> > ### Comment · Area_Chair_kw1z · 2024-08-12
> > **Has the rebuttal addressed your concerns?**
> >
> > Dear Reviewer dNLV,
> >
> > Thank you again for your time to review this paper. Could you please check if the authors' rebuttal has addressed your concerns at your earliest convenience? The deadline of the discussion period will end in about 24 hours. Thank you!
> >
> > Best regards,
> >
> > AC

---

> > ### Comment · Reviewer_dNLV · 2024-08-13
> >
> > Thanks for your responses. I will keep my rating.

---

> ### Author Response · Authors · 2024-08-13
>
> We thank the reviewer for getting back on the rebuttal. We would appreciate it if the reviewer could elaborate a bit more on what they think about the additional results and the discussions we provided in the rebuttal.
>
> Best,
> Authors

---

### Official Review · Reviewer_a22a · 2024-07-14

**Soundness:** 4
**Presentation:** 4
**Contribution:** 3
**Rating:** 7
**Confidence:** 5

**Summary:**

This paper studies the fine-grained understanding of objects, attributes, and relationships between objects for VLMs and MLLMs. Specifically, it focuses on the capability of VLMs and MLLMs to distinguish between two very similar images given a caption. Firstly, by leveraging an image generation diffusion model and an LLM, this paper introduces a new benchmark termed Visual Minimal-Change Understanding, which requires models to predict the correct image-caption match given two images and two captions. Secondly, it performs evaluations on multiple VLMs and MLLMs and analyzes the performance in terms of different types of changes: object, attribute, count, and spatial relation. Thirdly, it fine-tunes existing VLMs and MLLMs on the generated datasets and shows improvement in fine-grained understanding.

**Strengths:**

- The motivation is clear, and the problem addressed in this paper is well-explained.
- The proposed dataset is interesting and benefits the research of fine-grained understanding of foundational VLMs.
- This paper is well-written and easy to follow.
- The evaluation protocol and the methods for fine-tuning are technically sound.
- The analysis of VLM and MLLM performance on the proposed benchmark is interesting.

**Weaknesses:**

- Some notations are confusing, and implementation details for the evaluation are missing: how are two captions chosen in text scores for VLMs? It appears from line 247 that captions chosen from a single image are paired instead of being randomly sampled. What's the difference between C_0 and C_1 in lines 247 and 248? The notation here is a bit confusing. Do the notations C in line 247 and T in line 294 both refer to the captions for the images? What's the difference?


- Implementation details for fine-tuning are missing: How do the authors construct the VisMin Instruct-IT dataset? In line 344, some rule-based approaches are applied, but the reviewer cannot find details of these rule-based methods. It would be great to show some examples of how VisMin is converted to VisMin Instruct-IT. It also appears that the authors fine-tune the model using QALora and instruction fine-tuning techniques. How many instruction-image pairs are in this dataset?


- Insufficient dataset analysis: The authors mention in this paper that after human filtering of the proposed dataset, there is still some noise, such as deformation and mismatches. Conducting a human evaluation on the proposed dataset to determine the percentage of these noise data would be great. This would provide a valuable reference for future works using this dataset.

**Questions:**

- What is the motivation behind designing text scores and image scores for MLLMs? In Table 2, there is a significant drop in performance from text scores to image scores, especially for MLLMs. This drop might be because most MLLMs do not undergo instruction fine-tuning with two images, potentially affecting their ability to perform this task correctly. Under such circumstances, if the performance on image scores is low (for example, in spatial relation tasks), it becomes unclear whether the issue is the model's inability to process information from two images or its inability to detect minor changes in spatial relationships.
- Visual minimal changes occur frequently in videos. Do you have any experience in this direction, such as how this dataset and the fine-tuned model could benefit video understanding? (If it's discussed in the paper, could you point it out? I may have missed it.)
- Will the model and dataset be released? Also, VisMin Instruct-IT dataset.
- (Minor comment) GPT-4V has different versions; it would be great to specify which version was used in the paper.

**Limitations:**

- The authors discuss limitations and potential solutions in Section 7, such as the possibility of noisy data being included in the proposed dataset. The reviewer suggests conducting human evaluations on some samples from the dataset to provide statistics on the noisy data and to offer visualizations of these noisy data samples.
- The social impact should be discussed in this paper. This paper propose fine-tuned VLMs and MLLMs which are able to detect visual minimal-change and can have potential social impact.

---

> ### Author Rebuttal · Authors · 2024-08-07
>
> **Some notations are confusing**
>
> We apologize for this. The captions C_0 and C_1 are minimal-change pairs as described in Fig 1 of the review version. The C in L247 is exactly the same as T in L294. We will clarify these and make the notations consistent in the camera-ready version.
>
> **Details about Instruct-IT dataset**
>
> In the VisMin dataset, a sample consists of a source image-caption pair (I0,C0) and its minimally edited version (I1, C1). To construct the VisMin Instruct-IT dataset, we create four instruction samples from each sample in VisMin:
> * two samples for the Image-Task (selecting the best matching image given a caption): 1) given C0, select between I0, I1, 2) given C1, select between I0, I1
> * two samples for the Text-Task (selecting the best matching caption given an image): 1) given I0, select between C0, C1, 2) given I1, select between C0, C1
>
> Please refer to Table 3 in the rebuttal PDF for the exact set of instruction templates we used. We randomly sample one template for each task type when creating the instruction samples.
>
> In total, we create 65k instruction samples for training. Additionally, we include 16k randomly selected image-caption pairs (this could be either I0,C0 or I1,C1) to retain the general image captioning ability of the model.
>
> We will clarify these in the camera-ready version.
>
> **Possibility of noisy data being included in the proposed dataset, even after human filtering.**
>
> We believe the reviewer is referring to L376-378. These lines refer to our training set, not the human-filtered benchmark set. We will clarify this in the camera-ready version.
>
> **Human evaluations and statistics on the noisy data.**
>
> We did conduct a comprehensive four-stage human evaluation on the *entire* benchmark set as described in L174-188 of the review version (RV). The benchmark set is obtained after *filtering out the samples* that do not pass any of the four stages of human evaluation. We refer the reviewer to the RV for the evaluation criteria in each stage and the associated acceptance rates. All stages except the first stage have high acceptances rates (80-95%) indicating a low level of noise in our automatically generated data. For the first stage, the acceptance rate is 26%, due to a high rate of rejection (63%) of images that don’t look natural. We want to avoid unnatural looking images in the benchmark set to prevent models from easily recognizing such unnatural samples [1]. However, such noisy samples are acceptable in the training set as the goal is to teach minimal change understanding to the models. This is further validated by our experimental results where we see that finetuning on our noisy data improves model performance for both IID and OOD benchmarks.
>
> [1] Hsieh et al. SugarCrepe. NeurIPS Datasets and Benchmarks Track 2023.
>
> **Visualizations of these noisy data samples.**
>
> We have provided random samples from both training and benchmark splits in Fig 16 and 17 in the supplementary material.
>
> **Motivation behind designing text scores and image scores for MLLMs**
>
> Text and image scores are needed to disentangle a model’s capability of “distinguishing between two captions given an image” from “distinguishing between two images given a caption”. We use the same metric for both foundation VLMs and MLLMs to ensure **consistent** evaluation.
>
> **Drop from text scores to image scores in MLLMs due to lack of instruction fine-tuning with two images.**
>
> We agree with the concern about image scores and for this very reason, we chose Idefics2 for our finetuning experiments as it is capable of processing multiple images (it’s trained with multiple images). GPT4, InternVL and Gemini are also capable of processing multiple images. Thus, low image scores for these models suggest an inability to detect minimal changes. Also, if the low image score for a model was solely due to its inability to process multiple images, the model should perform poorly in all categories (Object, Attribute, S.Relation, Count), however we observe particularly low scores on some categories such as S.Relation and Count.
>
> **How could VisMin benefit video understanding**
>
> Our finetuned models excel in discerning semantic differences, but neighboring video frames often exhibit *low-level* rather than semantic changes. Abrupt changes between consecutive frames are rare, such as a school bus changing to a car, a shirt changing color, the number of people changing or the objects swapping their positions. Moreover, our dataset does not cover action changes which are very common in videos. Therefore, we expect finetuning on our dataset to result in limited improvements for nearby video frames. We quantified the extent of improvement by reporting model performance on three video subsets of the EQBen dataset that were created by sampling nearby frames (see Fig 1 of the rebuttal PDF for some examples). Results are presented in Table 2 of the rebuttal PDF. For both CLIP and Idefics2, finetuning on our minimal change data leads to significant improvements on EQ-YouCook2 (group score), and similar performance on the other two splits (group score). From our qualitative inspection of the samples, we find that EQ-YouCook2 has more significant object changes compared to other splits (which mainly involve action changes). Thus our dataset can benefit video understanding when it has the kinds of changes that our dataset covers.
>
>
> **Will the model and dataset be released?**
>
> Yes we will release all model weights, generated dataset as well as generation code.
>
> **GPT-4V version**
>
> We use GPT-4-turbo-2024-04-09.
>
> **The social impact should be discussed in this paper**
>
> Thanks, we will include this in the camera-ready version of the paper. We provide a brief discussion of this in the comments below.

---

> > ### Comment · Reviewer_a22a · 2024-08-12
> > **Response to rebuttal and further questions**
> >
> > Thank you for the further clarifications and the updated results/data. I appreciate the additional details and examples provided in the rebuttal PDF. The response has addressed most of my concerns. However, I would like to clarify a question I raised in my initial review. When I referred to an insufficient dataset analysis, I was specifically referring to the training set. The reason for this concern is that noisy training data used for fine-tuning MLLMs can lead to hallucinations during testing. Therefore, it would be beneficial for future work utilizing this training data to include a human evaluation on (a small subset of) the training set to determine the percentage of noisy data present. How noisy is the training set? Have the authors observed any hallucinations in testing? It would be greatly appreciated if the authors could provide a discussion or share their thoughts on this aspect.

---

> > > ### Author Response · Authors · 2024-08-12
> > > **Response to Follow-Up Questions from Reviewer a22a**
> > >
> > > Thank you for following up on our response. Please see our responses below inline.
> > >
> > > > Therefore, it would be beneficial for future work utilizing this training data to include a human evaluation on (a small subset of) the training set to determine the percentage of noisy data present. How noisy is the training set?
> > >
> > > The comprehensive four-stage human evaluation we conducted on the benchmark set *precisely* reflects the percentage of noise in the training data as the training set and the (unfiltered) benchmark set are IID with respect to each other. This is so because both sets are generated using the *exact same* automated data generation pipeline. The only difference is that the benchmark set undergoes human filtering (filtering out the samples that do not pass any of the four stages of human evaluation) while the training set does not.
> > >
> > > > Have the authors observed any hallucinations in testing? It would be greatly appreciated if the authors could provide a discussion or share their thoughts on this aspect.
> > >
> > > Thanks for the insightful question. We would like to share two insights here:
> > >
> > > * The POPE [1] benchmark we evaluated our Idefics2-Vismin model on is a benchmark specifically designed to evaluate *object hallucinations* in multimodal LLMs. Our results (figure 5 on page 8 of the review version) show that Idefics2-Vismin achieves comparable performance to Idefics2 (Idefics2: 85.5%, Idefics2-Vismin: 83.6%) suggesting that fine-tuning Idefics2 on our minimal-change training data did not significantly impact the degree of hallucinations in the base model.
> > >
> > > * The fine-grained understanding benchmarks we evaluate our fine-tuned models on (results presented in Tables 3 and 4 of the review version) assess a model's ability to correctly recognize precise objects, attributes, and counts of objects. We believe such evaluations are closely related to evaluating hallucinations in a model, particularly object (e.g., errors in object recognition and counting) and attribute hallucinations (e.g., inaccuracies in attribute recognition). The fact that our fine tuned models significantly improve the performance of the base models on *a suite of fine-grained understanding benchmarks*, spanning both IID and OOD benchmarks, suggests that fine-tuning on our proposed minimal change data does not increase hallucinations in the base model. If finetuning on our data increased hallucinations in the base model, we would expect to see performance degradation on the fine grained understanding benchmarks.
> > >
> > > [1] Li et al. Evaluating Object Hallucination in Large Vision-Language Models. EMNLP 2023.

---

> ### Author Response · Authors · 2024-08-07
> **Brief Discussion of Social Impact**
>
> The positive societal implications of our work are the same as those of any vision-langauge research (as we aim to make VLMs more accurate), such as aiding the visually impaired in understanding their surroundings, teaching children through interactive demos, and interacting with in-home physical robots. However, like most other technology, accurate vision-language systems could also be used for potentially harmful applications such as extracting private information from CCTV cameras deployed at public places, and leaking personal data like credit card details when used to assist visually impaired users. Robust privacy safeguards are crucial to mitigate these risks.

---

> ### Comment · Reviewer_a22a · 2024-08-12
> **Response to Follow-Up Comments form authors**
>
> Thank you for the response. My concern has been addressed in the discussion.
>
> After reviewing the feedback from other reviewers and the corresponding rebuttals, I believe this paper makes good contributions, including the data generation pipeline, the generated training and testing sets, and the further fine-tuned MLLM models. Additionally, the reviewer recognize that the task addressed in this paper has the potential for high impact, particularly given the current trend of extending MLLMs to handle multi-image and video data. In this context, the ability to recognize minimal changes is a fundamental capability for further perception and cognition tasks. Thus, I am glad to raise my score to "Accept," while remaining open to discussions from other reviewers. Lastly, I would like to remind the authors to include the results and discussion in the final version and to fulfill their commitment to releasing the data.

---

> > ### Author Response · Authors · 2024-08-12
> >
> > Thank you for recognizing that our paper has the potential for high impact! We really appreciate your encouraging comments. We will incorporate the additional analyses and discussion in the camera-ready version and will release the data.

---

### Official Review · Reviewer_Lkr8 · 2024-07-24

**Soundness:** 3
**Presentation:** 3
**Contribution:** 2
**Rating:** 5
**Confidence:** 4

**Summary:**

This paper introduces a new benchmark, VisMin, which mainly challenges models to detect semantic differences between visually similar but semantically different images. It uses an automated data curation pipeline and human verification to create dataset. The authors benchmark the dataset with current VLMs and MLLMs, showing that the proposed dataset can help improve image-text alignment and overall performance.

**Strengths:**

1. The motivation of the paper makes sense and the paper writing is clear.
2. The curated dataset contains hard negative samples that will be beneficial to improve fine-grained understanding of current models. The dataset curation pipeline is well designed and the data quality looks quite good.
3. It indicates huge performance on CLIP and Idefics2 after finetuning with the proposed dataset.

**Weaknesses:**

1. The paper only uses T, I, G to evaluate the performance of different models on the proposed dataset, which is insufficient to show the ability of understanding attribute, count and spatial relation. I think some basic metrics like accuracy are also needed.
2. The models being benchmarked are not sufficient. It lacks some latest or popular models, like Flava, CogVLM, etc.
3. Another concern is that the finetuned CLIP/Idefics2 models do not demonstrate improvements in all aspects, indicating that the benchmark may introduce some side effects. I am curious about the underlying reasons for this and recommend that the authors propose a new baseline method alongside the benchmark.

**Questions:**

1. I'd like to see evaluation results of more latest models, with more evaluation metrics.
2. I am also interested in seeing the authors provide justification for the finetuning results on the CLIP/Idefics2 models, since they are not consistently strong.

**Limitations:**

The authors use editing models to modify original images. I suggest that the authors carefully review the curated dataset to ensure there are no ethical issues.

---

> ### Author Rebuttal · Authors · 2024-08-07
>
> **Some basic metrics like accuracy are also needed.**
>
> The T (Text), I (Image), and G (Group) metrics we use *indeed measure accuracy*, i.e., the proportion of examples for which the model produces the correct output [1]. The correctness criteria is different for each of T, I, G, as stated in L244-255. To obtain the aggregate T, I, G scores for the entire test set, we average the per-sample T, I, G scores across all samples in the test set. Thus, the aggregate T, I, G scores represent the accuracies on the test set. We will clarify this in the camera-ready version.
>
> Moreover, these T, I, G metrics we use are standard metrics for evaluating fine-grained understanding, originally proposed in Winoground [2] and adopted in subsequent research efforts such as EQBEN [3] and SPEC [4].
>
> [1] Goodfellow et al. Deep Learning, Chapter 5. MIT Press 2016.
>
> [2] Thrush et al. Winoground. CVPR 2022.
>
> [3] Wang et al. Equivariant similarity for vision-language foundation models. ICCV 2023.
>
> [4] Peng et al. Synthesize, diagnose, and optimize. CVPR 2024.
>
> **The models being benchmarked are not sufficient.**
>
> We disagree with this point. Compared to the models suggested by the reviewer, such as Flava (2022) and CogVLM (2023), our selection *already* includes contemporary or more up-to-date models models like LLaVa1.6 (2023), Idefics2 (2024), InternVL1.5 (2023), GPT4V Turbo (2023), and Gemini1.5 Pro (2024). We chose representative models from various families – foundation VLMs and MLLMs, open-source and closed-source, covering 9 different models: CLIP (2021), SigLip (2023), BLIP (2022), Coca (2022), LLaVa1.6 (2023), Idefics2 (2024), InternVL1.5 (2023), GPT4V Turbo (2023 - closed source), and Gemini1.5 Pro (2024 - closed source). We believe our evaluation is extensive and up-to-date.
>
> As per R1’s request, we benchmarked Flava and CogVLM (results in Table 1 of the rebuttal PDF). CogVLM outperforms other MLLMs in understanding object and attribute changes, likely due to the incorporation of grounding tasks and high-quality human-annotated datasets like Flickr30K Entities [5], RefCOCO [6], and VisualGenome [7] in its pretraining. Flava performs similarly to other foundational VLMs, except for spatial relations where both FLAVA and BLIP outperform significantly. This is likely due to the incorporation of COCO [8]  in the pretraining datasets of these two models, which contains a significantly higher proportion of spatial reasoning data (10.74%), compared to web-scraped datasets such Conceptual Captions (5.72%) and LAION (2.32%) used to train other models benchmarked in the paper (see Figure 1 (right) in the PDF). To obtain the percentages, we compute the proportion of captions in each dataset that contain at least one of the following spatial relation words: left, right, top, bottom, below, above, under (these are all the spatial relation words in the VisMin spatial relation split).
>
> [5] Plummer et al. Flickr30K Entities. IJCV 2017.
>
>
> [6] Yu et al. Modeling Context in Referring Expressions. ECCV 2016.
>
>
> [7] Krishna et al. Visual Genome. IJCV 2017.
>
>
> [8] Lin et al. Microsoft COCO. ECCV 2014.
>
>
> **Reasons behind finetuned models not demonstrating improvements in all aspects.**
>
> We believe the reviewer is referring to the cases (listed below) where our finetuned models do not outperform the baselines. We *have already provided* the underlying reasons for each of these cases in the review version (RV), but we explain them again below:
> *  For spatial relation tasks in the IID setting (Table 3 in RV), we noted a slight performance degradation for CLIP upon fine-tuning on our minimal change data (L308-310). This is likely due to CLIP's contrastive loss design which often overlooks spatial prepositions, as also discussed in [9].
> * For OOD settings (Table 4 in RV), VisMin-CLIP *consistently outperforms the base CLIP* model but does not outperform NegCLIP on Sugarcrepe, Valse and ImageCode. For Sugarcrepe and Valse, this is likely due to the NegCLIP’s method of generating hard negatives, which uses linguistic variations similar to these benchmarks, making the evaluation more favorable for NegCLIP (L317-318). For ImageCode, this is likely because ImageCode requires understanding various challenging phenomena (L325-326), such as temporal markers, actions, negation, and visibility/occlusion (source: Table 2 of the ImageCode paper [10]) which our minimal change dataset does not cover.
>
> Additionally, for Idefics2, although finetuning on our data demonstrates overall improvements in both OOD and IID settings, there is a 2.8% decline in the performance on the VALSE benchmark. This is likely because VALSE tests for specific kinds of skills such as pronominal coreference resolution and understanding actions that our minimal change data does not cover.
>
> We would appreciate it if the reviewer could share their thoughts on these justifications we provided above.
>
> [9] Kamath et al. What's "up" with vision-language models?. EMNLP 2023.
>
>
> [10] Krojer et al. Image Retrieval from Contextual Descriptions. ACL 2022.
>
> **“propose a new baseline method alongside the benchmark.”**
>
> Our paper aims to benchmark existing models and investigate if fine-tuning on minimally changed data alone can boost performance, *without introducing new methods or losses*. We propose a *model-agnostic* recipe to improve fine-grained understanding capabilities of both foundational VLMs and multimodal LLMs, which we believe would be of sufficient value to the community.
>
> **Potential ethical issues in the dataset**
>
> Since we only make minimal edits (object, attribute, counting, spatial relations) on top of published publicly available datasets such as COCO, we do not anticipate our edited images to have any data privacy, copyright, and consent issues following the definitions in the NeurIPS Ethics Guidelines. See more details in the comments below.

---

> ### Author Response · Authors · 2024-08-07
> **Addressing Potential Ethical Issues in the Dataset**
>
> Moreover, for the images we generated from scratch (counting and spatial relations), as well as the images edited on top of COCO, we avoid using Personally Identifiable Information (PII) in our text prompts used for generation (e.g., use common nouns such "a girl" instead of names entities such as "taylor swift"). We also filter all the edited images using an Not Safe For Work (NSFW) filter.

---

> > ### Comment · Area_Chair_kw1z · 2024-08-12
> > **Has the rebuttal addressed your concerns?**
> >
> > Dear Reviewer Lkr8,
> >
> > Thank you again for your time to review this paper. Could you please check if the authors' rebuttal has addressed your concerns at your earliest convenience? The deadline of the discussion period will end in about 24 hours. Thank you!
> >
> > Best regards,
> >
> > AC

---

> > > ### Comment · Reviewer_Lkr8 · 2024-08-14
> > >
> > > Thank you. The author has addressed most of my concerns. I raised my score to 5, because I think ideally a model-agnostic recipe paper should demonstrate improvements in all aspects.

---

> > > > ### Author Response · Authors · 2024-08-14
> > > >
> > > > We thank the reviewer for raising the score and for providing their thoughts. However, we would like to note that our proposed recipe (finetuning of base VLMs with our proposed minimal change data) achieves the best performance, outperforming both the base models and existing methods (in the case of CLIP), in 25 out of 36 cases spanning 2 base models (CLIP and Idefics2), and multiple benchmarks including both IID and OOD benchmarks, both fine-grained understanding and standard image-text understanding benchmarks. We respectfully disagree with the reviewer that a model-agnostic recipe should demonstrate improvements in *all* cases for it to be a useful contribution for the research community.
> > > >
> > > > Best,
> > > > Authors

---

### Author Rebuttal · Authors · 2024-08-07

We thank the reviewers for their thoughtful feedback! We’re encouraged that they found the **motivation** of the paper to be **clear** (R1, R2), the **proposed idea** of using minimal change pairs to improve fine-grained understanding **promising** (R3), the **data curation pipeline** to be **well designed** (R1), and our **curated data** to be of **quite good quality** (R1), and **beneficial for improving fine-grained understanding** (R2). Further, the reviewers acknowledged that the proposed benchmark **reveals significant shortcomings** of current vision-language models (R3) and that fine-tuning these models on the proposed minimal change data results in **huge performance improvements** (R1). The reviewers also appreciated the **technically sound** evaluation protocol and fine-tuning methods (R2), and the **interesting analysis** of VLM and MLLM performance presented in the paper (R2). . Finally, we are elated that the reviewers found our **paper writing** to be **clear**, **well-written**, and **easy to follow** (R1, R2).

**We believe we have addressed all of the concerns of each reviewer. We hope they would consider increasing their respective scores.**

---

### Comment · Area_Chair_kw1z · 2024-08-09

Dear Reviewers,

Thank you very much again for your valuable service to the NeurIPS community.

As the authors have provided detailed responses, it would be great if you could check them and see if your concerns have been addressed. Your prompt feedback would provide an opportunity for the authors to offer additional clarifications if needed.

Best regards,

AC

---

### Author Response · Authors · 2024-08-11
**Follow-Up on Our Response and Request for Score Reconsideration**

Dear Reviewers,

We hope this message finds you well. We are writing to follow up on the explanations we provided in response to your valuable feedback. We genuinely appreciate the time and effort you’ve invested in reviewing our work. If our clarifications have adequately addressed your concerns, we would kindly ask you to reconsider your score.

We sincerely hope that our work, which introduces the VisMin benchmark to challenge models in detecting fine-grained differences, develops an automated pipeline for high-quality data creation, and enhances fine-grained understanding in VLMs through fine-tuning, will be recognized as a valuable contribution to the community.

As August 13th marks the end of the discussion period, we would be grateful if you could share any additional questions or concerns you may have before then, so we can address them in a timely manner. Thank you once again for your thoughtful review.



Best regards,

Authors

---

### Decision · Program_Chairs · 2024-09-25

**Decision:**

Accept (poster)

**Comment:**

This paper received overall review scores, where most of the concerns of reviewers' concerns have been addressed after the rebuttal. Specifically, the reviewers acknowledge the clear motivation of explanation of the proposed approach, significance and impact of the proposed dataset, and the analysis of VLM and MLLM performance on the proposed benchmark.

The AC also found the authors' response to the ethical reviews convincing. So the AC recommends to accept this paper as a Poster.